# Hierarchical Reinforcement Learning with Uncertainty-Guided Diffusional Subgoals

**Vivienne Huiling Wang** [* 1]    **Tinghuai Wang** [* 2]    **Joni Pajarinen** [1]

## Abstract

Hierarchical reinforcement learning (HRL) learns to make decisions on multiple levels of temporal abstraction. A key challenge in HRL is that the low-level policy changes over time, making it difficult for the high-level policy to generate effective subgoals. To address this issue, the high-level policy must capture a complex subgoal distribution while also accounting for uncertainty in its estimates. We propose an approach that trains a conditional diffusion model regularized by a Gaussian Process (GP) prior to generate a complex variety of subgoals while leveraging principled GP uncertainty quantification. Building on this framework, we develop a strategy that selects subgoals from both the diffusion policy and GP's predictive mean. Our approach outperforms prior HRL methods in both sample efficiency and performance on challenging continuous control benchmarks.

## 1. Introduction

In the domain of Hierarchical Reinforcement Learning (HRL), the strategy to simplify complex decision-making processes involves structuring tasks into various levels of temporal and behavioral abstractions. This strategy excels particularly in environments where the challenges include long-term credit assignment and sparse rewards, making it a promising solution for long-horizon decision-making problems. Among the prevailing HRL paradigms, goal-conditioned HRL has been extensively explored in various studies (Dayan & Hinton, 1992; Schmidhuber & Wahnsiedler, 1993; Kulkarni et al., 2016; Vezhnevets et al., 2017; Nachum et al., 2018; Levy et al., 2019; Zhang et al., 2020;

*Equal contribution [1]Department of Electrical Engineering and Automation, Aalto University, Finland [2]Huawei Helsinki Research Center, Finland. Correspondence to: Vivienne Huiling Wang <vivienne.wang@aalto.fi>.

*Proceedings of the 42nd International Conference on Machine Learning*, Vancouver, Canada. PMLR 267, 2025. Copyright 2025 by the author(s).

Li et al., 2021; Kim et al., 2021; Li et al., 2023; Wang et al., 2023a; 2024). Within this framework, a high-level policy decomposes the primary goal into a series of subgoals, effectively directing the lower-level policy to achieve these subgoals. The efficacy of this approach hinges on the ability to generate subgoals that are semantically coherent and achievable, providing a strong learning signal for the lower-level policies. The hierarchical structure not only enhances the learning process's efficiency but also considerably improves the policy's overall performance in solving complex tasks.

The challenge with off-policy HRL lies in the simultaneous training of both high-level and low-level policies, which can lead to a changing low-level policy that causes past experiences in reaching previously achievable subgoals to become invalid. This issue requires the high-level policy to swiftly adapt its strategy to generate subgoals that align with the constantly shifting low-level skills. Previous works like HIRO (Nachum et al., 2018) and HAC (Levy et al., 2019) have attempted to address this problem through a relabeling strategy that utilizes hindsight experience replay (HER) (Andrychowicz et al., 2017a). This involves relabeling past experiences with high-level actions, *i.e.*, subgoals, that maximize the probability of the past lower-level actions. Essentially, the subgoal that induced a low-level behavior in the past experience is relabeled so that it potentially induce the similar low-level behavior with the current low-level policy. However, the relabeling approach does not facilitate efficient training of the high-level policy to comply promptly with updates to the low-level policy. Some studies (Zhang et al., 2020; Kim et al., 2021) have proposed that the problem of inefficient training is aggravated by the size of the subgoal space, and attempted to constrain it. While such constraints can improve performance in specific settings, they often fail to scale effectively to more complex environments.

To address these challenges, this work introduces a conditional diffusion model-based approach for subgoal generation (Sohl-Dickstein et al., 2015; Song et al., 2021; Ho et al., 2020). By directly modeling a highly expressive state-conditioned distribution of subgoals, our method offers a promising alternative to temporal difference learning, prov-

ing less susceptible to the destabilizing interactions inherent in the deadly triad (Sutton & Barto, 2018) and more adept at handling stochastic environments. While diffusion models excel at generating complex distributions, they often demand substantial training data and lack explicit uncertainty quantification. To mitigate these limitations and regularize learning, we incorporate a Gaussian Process (GP) prior as an explicit surrogate distribution for subgoals. The GP prior facilitates more efficient learning of the diffusion model based subgoal generation while simultaneously providing uncertainty quantification, informing the diffusion process about uncertain state regions.

Building upon this foundation, we further develop a subgoal selection strategy that combines the GP's predictive mean with subgoals sampled from a diffusion model, leveraging their complementary strengths. The GP's predictive mean, informed by state-subgoal pairs in the high-level replay buffer, aligns subgoals with feasible trajectories and ensures consistency with the underlying structure of the state space. This is achieved through the GP's kernel, which captures correlations between states and subgoals based on patterns learned from the training data. To complement this precision, the diffusion model introduces adaptability by generating diverse subgoals from its learned distribution. Together, these approaches form a hybrid strategy: the GP offers structured, data-driven guidance towards reliable, low-uncertainty subgoals, while the diffusion model provides flexibility, enabling robust subgoal generation that balances reliability with versatility.

Our main contributions are summarized as follows:

- We introduce a conditional diffusion model for subgoal generation, directly modeling a state-conditioned distribution and reducing susceptibility to instability from temporal difference learning.

- We employ a Gaussian Process (GP) prior to regularize the diffusion model and explicitly quantify uncertainty, promoting more efficient learning and reliable subgoal generation.

- We introduce a subgoal selection strategy that combines the GP's predictive mean, which aligns subgoals with feasible trajectories and ensures structural consistency, with the diffusion model's expressiveness, resulting in robust and adaptive subgoal generation.

## 2. Preliminaries

**Goal-conditioned HRL**   In reinforcement learning (RL), agent-environment interactions are modeled as a Markov Decision Process (MDP) denoted by $M = \langle \mathcal{S}, \mathcal{A}, \mathcal{P}, \mathcal{R}, \gamma \rangle$, where $\mathcal{S}$ represents the state space, $\mathcal{A}$ denotes the action

set, $\mathcal{P} : \mathcal{S} \times \mathcal{A} \times \mathcal{S} \to [0, 1]$ is the state transition probability function, $\mathcal{R} : \mathcal{S} \times \mathcal{A} \to \mathbb{R}$ is the reward function, and $\gamma \in [0, 1)$ signifies the discount factor. A stochastic policy $\pi(a|s)$ maps any given state $s$ to a probability distribution over the action space, with the goal of maximizing the expected cumulative discounted reward $\mathbb{E}_\pi[\sum_{t=0}^{\infty} \gamma^t r_t]$, where $r_t$ is the reward received at discrete time step $t$.

In a continuous control RL setting, modeled as a finite-horizon, goal-conditioned MDP $M = \langle \mathcal{S}, \mathcal{G}, \mathcal{A}, \mathcal{P}, \mathcal{R}, \gamma \rangle$, where $\mathcal{G}$ represents a set of goals, we employ a Hierarchical Reinforcement Learning (HRL) framework comprising two layers of policy akin to (Nachum et al., 2018). This framework includes a high-level policy $\pi_h(g|s)$ that generates a high-level action, or subgoal, $g_t \sim \pi_h(\cdot|s_t) \in \mathcal{G}$, every $k$ time steps when $t \equiv 0 \pmod k$. Between these intervals, a predefined goal transition function $g_t = f(g_{t-1}, s_{t-1}, s_t)$ is applied when $t \not\equiv 0 \pmod k$. The high-level policy influences the low-level policy through intrinsic rewards for achieving these subgoals. Following prior work (Andrychowicz et al., 2017a; Nachum et al., 2018; Zhang et al., 2020; Kim et al., 2021), we define the goal set $\mathcal{G}$ as a subset of the state space, *i.e.*, $\mathcal{G} \subset \mathcal{S}$, and the goal transition function as $f(g_{t-1}, s_{t-1}, s_t) = s_{t-1} + g_{t-1} - s_t$. The objective of the high-level policy is to maximize the extrinsic reward as defined by $r_t^h = \sum_{i=t}^{t+k-1} R_i, \quad t = 0, 1, 2, \ldots,$ where $R_i$ is the environmental reward.

The low-level policy aims to maximize the intrinsic reward granted by the high-level policy. It accepts the high-level action or subgoal $g$ as input, interacting with the environment by selecting an action $a_t \sim \pi_l(\cdot|s_t, g_t) \in \mathcal{A}$ at each time step. An intrinsic reward function, $r_t^l = -\|s_t + g_t - s_{t+1}\|_2$, evaluates the performance in reaching the subgoal $g$.

This goal-conditioned HRL framework facilitates early learning signals for the low-level policy even before a proficient goal-reaching capability is developed, enabling concurrent end-to-end training of both high and low-level policies. However, off-policy training within this HRL framework encounters the non-stationarity problem as highlighted in Section 1. HIRO (Nachum et al., 2018) addresses this by relabeling high-level transitions $(s_t, g_t, \sum_{i=t}^{t+k-1} R_i, s_{t+k})$ with an alternate subgoal $\tilde{g}_t$ to enhance the likelihood of the observed low-level action sequence under the current low-level policy, by maximizing $\pi_l(a_{t:t+k-1}|s_{t:t+k-1}, \tilde{g}_{t:t+k-1})$.

The relabeled subgoals are generally considered to be sampled from a distribution that asymptotically approximates an optimal high-level policy within a stationary data distribution (Zhang et al., 2020; Wang et al., 2023a). This leads to the conjecture that learning a conditional distribution based on the relabeled subgoals inherently facilitates the achievement of stationarity in the high-level policy.

**Diffusion Model** Diffusion models (Sohl-Dickstein et al., 2015; Song et al., 2021; Ho et al., 2020) have emerged as a powerful framework for generating complex data distributions. These models pose the data-generating process as an iterative denoising procedure $p_\theta(\mathbf{x}_{t-1}|\mathbf{x}_t)$. This denoising is the reverse of a diffusion or forward process which maps a data example $\mathbf{x}_0 \sim q(\mathbf{x}_0)$ through a series of intermediate variables $\mathbf{x}_{1:T}$ in $T$ steps with a pre-defined variance schedule $\beta_i$, according to

$$q(\mathbf{x}_{1:T}|\mathbf{x}_0) := \prod_{t=1}^{T} q(\mathbf{x}_t|\mathbf{x}_{t-1}), \tag{1}$$

$$q(\mathbf{x}_t|\mathbf{x}_{t-1}) := \mathcal{N}(\mathbf{x}_t; \sqrt{1-\beta_t}\mathbf{x}_{t-1}, \beta_t\mathbf{I}). \tag{2}$$

The reverse process is constructed as $p_\theta(\mathbf{x}_{0:T}) := \mathcal{N}(\mathbf{x}_T; 0, \mathbf{I}) \prod_{t=1}^{T} p_\theta(\mathbf{x}_{t-1}|\mathbf{x}_t)$. Parameters $\theta$ are optimized by maximizing a variational bound on the log likelihood of the reverse process $\mathcal{L}_{\text{ELBO}} = \mathbb{E}_{q(\mathbf{x}_{1:T}|\mathbf{x}_0)}[\log \frac{p_\theta(\mathbf{x}_{1:T}|\mathbf{x}_0)}{q(\mathbf{x}_{1:T}|\mathbf{x}_0)}]$.

In this paper, we employ two distinct notations of timesteps: one for the diffusion process and another for the reinforcement learning trajectory. Specifically, we denote the diffusion timesteps with superscripts $i \in \{1, \ldots, N\}$, and the trajectory timesteps with subscripts $t \in \{1, \ldots, T\}$.

## 3. Proposed Method

In this section, we present our method — **HI**erarchical RL subgoal generation with **DI**ffusion model (HIDI), which explicitly models the state-conditional distribution of subgoals at the higher level. By combining with the temporal difference learning objective, the high-level policy promptly adapts to generating subgoals following a data distribution compatible with the current low-level policy.

### 3.1. Diffusional Subgoals

We formulate the high-level policy as the reverse process of a conditional diffusion model as

$$\pi_{\theta_h}^h(\boldsymbol{g}|\boldsymbol{s}) := p_{\theta_h}(\boldsymbol{g}^{0:N}|\boldsymbol{s}) = \mathcal{N}(\boldsymbol{g}^N; \mathbf{0}, \boldsymbol{I}) \prod_{i=1}^{N} p_{\theta_h}(\boldsymbol{g}^{i-1}|\boldsymbol{g}^i, \boldsymbol{s}),$$

with the end sample $\boldsymbol{g}_i$ being the generated subgoal. Following Ho et al. (2020), a Gaussian distribution $N(\boldsymbol{g}^{i-1}; \mu_{\theta_h}(\boldsymbol{g}^i, \boldsymbol{s}, i), \Sigma_{\theta_h}(\boldsymbol{g}^i, \boldsymbol{s}, i))$ is used to model $p_{\theta_h}(\boldsymbol{g}^{i-1}|\boldsymbol{g}^i, \boldsymbol{s})$, which is parameterized as a noise prediction model with learnable mean

$$\boldsymbol{\mu}_{\theta_h}\left(\boldsymbol{g}^i, \boldsymbol{s}, i\right) = \frac{1}{\sqrt{\alpha_i}}\left(\boldsymbol{g}^i - \frac{\beta_i}{\sqrt{1-\bar{\alpha}_i}}\boldsymbol{\epsilon}_{\theta_h}\left(\boldsymbol{g}^i, \boldsymbol{s}, i\right)\right)$$

and fixed covariance matrix $\boldsymbol{\Sigma}_{\theta_h}\left(\boldsymbol{g}^i, \boldsymbol{s}, i\right) = \beta_i \boldsymbol{I}$.

Starting with Gaussian noise, subgoals are then iteratively generated through a series of reverse denoising steps by the noise prediction model parameterized by $\theta_h$ as

$$\boldsymbol{g}^{i-1} = \frac{1}{\sqrt{\alpha_i}}\left(\boldsymbol{g}^i - \frac{\beta_i}{1-\bar{\alpha}_i}\boldsymbol{\epsilon}_{\theta_h}(\boldsymbol{g}^i, s, i)\right) + \sqrt{\beta_i}\boldsymbol{\epsilon},$$

$$\boldsymbol{\epsilon} \sim \mathcal{N}(\mathbf{0}, \boldsymbol{I}) \text{ if } i > 1, \text{ else } \boldsymbol{\epsilon} = 0. \tag{3}$$

The learning objective of our subgoal generation model comprises three terms, *i.e.*, diffusion objective $\mathcal{L}_{dm}(\theta_h)$, GP based uncertainty $\mathcal{L}_{gp}(\theta_h, \theta_{gp})$ and RL objective $\mathcal{L}_{dpg}(\theta_h)$:

$$\pi_h = \operatorname*{argmin}_{\theta_h} \mathcal{L}_d(\theta_h) := \mathcal{L}_{dm}(\theta_h) + \psi\mathcal{L}_{gp}(\theta_h, \theta_{gp})$$

$$+ \eta\mathcal{L}_{dpg}(\theta_h), \tag{4}$$

where $\psi$ and $\eta$ are hyperparameters.

We adopt the objective proposed by Ho et al. (2020) as the diffusion objective,

$$\mathcal{L}_{dm}^h(\theta_h) = \mathbb{E}_{i\sim\mathcal{U},\epsilon\sim\mathcal{N}(\mathbf{0},\boldsymbol{I}),(\boldsymbol{s},\boldsymbol{g})\sim\mathcal{D}_h} \tag{5}$$

$$\left[\left\|\boldsymbol{\epsilon} - \boldsymbol{\epsilon}_{\theta_h}\left(\sqrt{\bar{\alpha}_i}\boldsymbol{g} + \sqrt{1-\bar{\alpha}_i}\epsilon, \boldsymbol{s}, i\right)\right\|^2\right], \tag{6}$$

where $\mathcal{D}_h$ is the high-level replay buffer, with the subgoals relabeled similarly to HIRO. Specifically, relabeling $\boldsymbol{g}_t$ in the high-level transition $(\boldsymbol{s}_t, \boldsymbol{g}_t, \sum_{i=t}^{t+k-1} R_i, \boldsymbol{s}_{t+k})$ with $\tilde{\boldsymbol{g}}_t$ aims to maximize the probability of the incurred low-level action sequence $\pi_l(\boldsymbol{a}_{t:t+k-1}|\boldsymbol{s}_{t:t+k-1}, \tilde{\boldsymbol{g}}_{t:t+k-1})$, which is approximated by maximizing the log probability

$$\log \pi_l\left(\boldsymbol{a}_{t:t+k-1} \mid \boldsymbol{s}_{t:t+k-1}, \tilde{\boldsymbol{g}}_{t:t+k-1}\right)$$

$$\propto -\frac{1}{2} \sum_{i=t}^{t+k-1} \left\|\boldsymbol{a}_i - \pi_{\theta_l}^l(\boldsymbol{s}_i, \tilde{\boldsymbol{g}}_i)\right\|_2^2 + \text{const.} \tag{7}$$

The purpose of the above diffusion objective is to align the high-level policy's behavior with the distribution of "optimal" relabeled subgoals, thereby mitigating non-stationarity in hierarchical models.

While our method is versatile enough to be integrated with various actor-critic based HRL frameworks, we specifically utilize the TD3 algorithm (Fujimoto et al., 2018) at each hierarchical level, in line with HIRO (Nachum et al., 2018), HRAC (Zhang et al., 2020) and HIGL (Kim et al., 2021). Accordingly, the primary goal of the subgoal generator within this structured approach is to optimize for the maximum expected return as delineated by a deterministic policy. This objective is formulated as:

$$\mathcal{L}_{dpg}(\theta_h) = -\mathbb{E}_{\mathbf{s}\sim\mathcal{D}_h,\mathbf{g}^0\sim\pi_{\theta_h}}\left[Q_h(\mathbf{s}, \mathbf{g}^0)\right]. \tag{8}$$

Given that $\mathbf{g}^0$ (hereafter referred to as **g** without loss of generality) is reparameterized according to Eq. 3, the gradient of $\mathcal{L}_{dpg}$ with respect to the subgoal can be backpropagated through the entire diffusion process.

## 3.2. Uncertainty Modeling with Gaussian Process Prior

Whilst diffusion models possess sufficient expressivity to model the conditional distribution of subgoals, they face two significant challenges in the context of subgoal generation in HRL. Firstly, these models typically require a substantial amount of training data, *i.e.*, relabeled subgoals, to achieve accurate and stable performance, which can be highly sample inefficient. Secondly, standard diffusion models lack an inherent mechanism for quantifying uncertainty in their predictions, a crucial aspect for effective exploration and robust decision-making in HRL. To address these issues, we propose to model the state-conditional distribution of subgoals at the high level, harnessing a Gaussian Process (GP) prior as an explicit surrogate distribution for subgoals.

Given $\mathcal{D}_h$, the relabeled high-level replay buffer, a zero-mean Gaussian process prior is placed on the underlying latent function, i.e., the high-level policy, $\mathbf{g} \sim \pi_{\theta_h}$, to be modeled. This results in a multivariate Gaussian distribution over any finite subset of latent variables:

$$p(\mathbf{g}|\mathbf{s}; \theta_{gp}) = \mathcal{N}(\mathbf{g}|\mathbf{0}, \mathbf{K}_N + \sigma^2 \mathbf{I}), \qquad (9)$$

where the covariance matrix $\mathbf{K}_N$ is constructed from a covariance function, or kernel, which expresses a prior notion of smoothness of the underlying function: $[\mathbf{K}_N]_{ij} = K(\mathbf{s}_i, \mathbf{s}_j)$. Typically, the covariance function depends on a small number of hyperparameters $\theta_{gp}$, which control these smoothness properties. Without loss of generality, we employ the commonly used Radial Basis Function (RBF) kernel:

$$K(\mathbf{s}_i, \mathbf{s}_j) = \gamma^2 \exp\left[-\frac{1}{2\ell^2} \sum_{d=1}^{D} \left(s_i^{(d)} - s_j^{(d)}\right)^2\right],$$

$$\theta_{gp} = \{\gamma, \ell, \sigma\}. \qquad (10)$$

Here, $D$ is the state space dimension, $\gamma^2$ is the variance parameter, $\ell$ is the length scale parameter, and $\sigma^2$ is the noise variance. The hyperparameters $\theta_{gp}$ are learnable parameters of the GP model.

We leverage the GP prior to regularize and guide the optimization of the diffusion policy. Specifically, we incorporate the negative log marginal likelihood of the GP as an additional loss term $\mathcal{L}_{gp}$:

$$\mathcal{L}_{gp} = \mathbb{E}_{\mathbf{s} \sim \mathcal{D}_h, \mathbf{g} \sim \pi_{\theta_h}} \left[-\log p(\mathbf{g}|\mathbf{s}; \theta_h, \theta_{gp})\right]$$

$$= \mathbb{E}_{\mathbf{s} \sim \mathcal{D}_h, \mathbf{g} \sim \pi_{\theta_h}} \left[-\frac{1}{2} \mathbf{g}^\top (\mathbf{K}_N + \sigma^2 \mathbf{I})^{-1} \mathbf{g}\right.$$

$$\left. -\frac{1}{2} \log |\mathbf{K}_N + \sigma^2 \mathbf{I}| - \frac{N}{2} \log 2\pi\right]. \qquad (11)$$

By minimizing $\mathcal{L}_{gp}$ alongside the diffusion loss and RL objective, we encourage the diffusion policy to generate sub-

goals that are consistent with the GP prior, particularly by focusing learning on feasible regions supported by previously achieved transitions. This approach not only regularizes the diffusion model but also incorporates the uncertainty estimates provided by the GP, potentially leading to more robust and sample-efficient learning.

To formalize the effect of the GP regularization on the diffusion-based subgoal policy, we provide the following theorem and proposition. The GP loss introduces a gradient signal that guides subgoal generation toward reliable regions identified by the GP posterior. Specifically:

**Theorem 3.1** (GP Regularization Guides Subgoal Alignment). *Let $\mathbf{g} = f(\boldsymbol{\epsilon}', \mathbf{s}; \theta_h)$ be the subgoal generated by the diffusion model conditioned on state $\mathbf{s}$ and noise $\boldsymbol{\epsilon}'$. Under mild regularity assumptions, the GP regularization term in the high-level objective encourages the learned distribution $p_{\theta_h}(\mathbf{g}|\mathbf{s})$ to align with the GP predictive mean $\mu_*(\mathbf{s})$ in regions of low predictive variance $\sigma_*^2(\mathbf{s})$.*

We further quantify the mechanism through which this alignment occurs:

**Proposition 3.2** (Gradient Weighting by GP Uncertainty). *Let $\mathbf{g} = f(\boldsymbol{\epsilon}', \mathbf{s}; \theta_h)$ be the subgoal generated by the diffusion model using the reparameterization trick. Then the gradient of the GP loss with respect to $\theta_h$ satisfies:*

$$\nabla_{\theta_h} L_{gp} = \mathbb{E}_{\mathbf{s}, \boldsymbol{\epsilon}'} \left[\left(\frac{\mathbf{g} - \mu_*(\mathbf{s})}{\sigma_*^2(\mathbf{s})}\right)^\top \nabla_{\theta_h} \mathbf{g}\right],$$

*where $\mu_*(\mathbf{s})$ and $\sigma_*^2(\mathbf{s})$ denote the GP predictive mean and variance. This implies that the GP regularization applies stronger gradient pressure for parameter updates in state regions $\mathbf{s}$ where the GP is more confident (i.e., $\sigma_*^2(\mathbf{s})$ is small).*

The detailed derivation and full proofs are provided in Appendix A.1.3.

While the GP framework provides explicit uncertainty quantification, standard GP training requires $\mathcal{O}(N^3)$ computation to invert the covariance matrix, which is infeasible for large replay buffers ($N$). To improve efficiency, we adopt a sparse GP approach using $M \ll N$ inducing states $\bar{\mathbf{s}}$ (Snelson & Ghahramani, 2005). These inducing states act as a summary of the full dataset $\mathcal{D}_h$.

Given a new state $\mathbf{s}_*$, the sparse GP yields a predictive distribution for the subgoal $\mathbf{g}_*$:

$$p(\mathbf{g}_*|\mathbf{s}_*, \mathbf{D}_h, \bar{\mathbf{s}}) = \mathcal{N}(\mathbf{g}_* \mid \mu_*, \sigma_*^2). \qquad (12)$$

where the predictive mean $\mu_*$ and variance $\sigma_*^2$ are given by:

$$\mu_* = \mathbf{k}_*^\top \mathbf{Q}_M^{-1} \mathbf{K}_{MN} (\Lambda + \sigma^2 \mathbf{I})^{-1} \mathbf{g} \qquad (13)$$

and

$$\sigma_*^2 = K_{**} - \mathbf{k}_*^\top (\mathbf{K}_M^{-1} - \mathbf{Q}_M^{-1})\mathbf{k}_* + \sigma^2. \qquad (14)$$

Here, $\mathbf{k}_*$, $\mathbf{Q}_M$, $\mathbf{K}_{MN}$, $\Lambda$, $\mathbf{K}_M$, and $K_{**}$ depend on the kernel function evaluated at the inducing states $\bar{\mathbf{s}}$, the query state $\mathbf{s}_*$, and the states in the replay buffer $\mathcal{D}_h$. The inducing states $\bar{\mathbf{s}}$ and GP hyperparameters $\theta_{gp}$ are learned by maximizing the marginal likelihood. The detailed derivation of the sparse GP is provided in Appendix A.3. The learned inducing states implicitly define regions of interest, informing the subgoal generation process.

### 3.3. Inducing States Informed Subgoal Selection

Building on the predictive distribution of the sparse GP model, we develop a subgoal selection strategy that primarily relies on subgoals sampled from a diffusion model, with the GP's predictive mean acting as a complementary regularizer. At each high-level decision point, subgoals are selected using a hybrid mechanism that integrates the expressive capabilities of the diffusion model with the structural guidance provided by the GP.

Subgoals are predominantly sampled from the diffusion policy $\pi_{\theta_h}(\mathbf{g} \,|\, \mathbf{s}_*)$, which generates diverse and adaptive subgoals from its learned distribution. Complementing this, the GP's predictive mean, $\mu_*$ (Eq. 13), derived from the sparse GP with inducing states and informed by state-subgoal pairs in the high-level replay buffer, leverages its kernel to capture smoothness and correlations in the state space. This regularizes subgoal selection by encouraging consistency with the structure and dynamics observed in the training data, anchoring the selection process to meaningful patterns *i.e.*, low-uncertainty regions supported by experience, while mitigating over-reliance on the diffusion model's flexibility.

The selection strategy is formalized as:

$$\mathbf{g}_* = \begin{cases} \boldsymbol{\mu}_*, & \text{with probability } \varepsilon, \\ \mathbf{g} \sim \pi_{\theta_h}(\mathbf{g} \,|\, \mathbf{s}_*), & \text{with probability } 1 - \varepsilon. \end{cases} \qquad (15)$$

To quantify the benefit of this subgoal selection, we consider the single macro-step reward $R(\mathbf{s}, \mathbf{g})$ for taking subgoal $\mathbf{g} \in \mathcal{G}$ in state $\mathbf{s}$, defined as:

$$R(\mathbf{s}, \mathbf{g}) \;=\; \mathbb{E}\Big[\sum_{i=0}^{k-1} r_{t+i} \,\Big|\, s_t = \mathbf{s}, \, g_t = \mathbf{g}\Big],$$

where $k$ is the number of lower-level steps that attempt to reach subgoal $\mathbf{g}$.

Under a near-optimal diffusion policy assumption (inspired by similar assumptions in actor-critic analysis (Kakade & Langford, 2002)), we have the following guarantee for the regret of our subgoal selection strategy:

**Theorem 3.3** (Single-Step Regret Bound for the Subgoal Selection). *Let $R^*(\mathbf{s}) = \max_{\mathbf{g} \in \mathcal{G}} R(\mathbf{s}, \mathbf{g})$. Suppose there is a baseline reward $R_{\min}$ such that for all $\mathbf{s} \in \mathcal{S}$, $R\big(\mathbf{s}, \boldsymbol{\mu}(\mathbf{s})\big) \geq R_{\min}$. Under the assumption of a near-optimal diffusion policy (see Appendix A.2 for the full statement), the subgoal selection strategy defined above has a bounded single-step regret.*

Furthermore, under standard policy improvement assumptions (similar to those in policy gradient methods (Sutton & Barto, 2018)), incorporating the GP mean provides a theoretical guarantee for non-decreasing performance:

**Proposition 3.4** (Single-Step Policy Improvement). *Let $Q_h(\mathbf{s}, \mathbf{g})$ be the high-level Q-value. Under certain conditions (see Appendix A.2 for the full statement), selecting the GP mean subgoal with a certain probability does not degrade performance in a single-step sense.*

The detailed proofs for Theorem 3.3 and Proposition 3.4 are provided in Appendix A.2 and A.2, respectively.

## 4. Related Work

HRL has been a significant field of study for addressing challenges such as long-term credit assignment and sparse rewards. It typically involves a high-level policy that breaks down the overarching task into manageable subtasks, which are then addressed by a more specialized low-level policy (Dayan & Hinton, 1992; Schmidhuber & Wahnsiedler, 1993; Kulkarni et al., 2016; Vezhnevets et al., 2017; Nachum et al., 2018; Levy et al., 2019; Zhang et al., 2020; Li et al., 2021; Kim et al., 2021; Li et al., 2023; Wang et al., 2023a; 2024). The mechanism of this decomposition varies, with some approaches utilizing discrete values for option or skill selection (Bacon et al., 2017; Fox et al., 2017; Gregor et al., 2017; Konidaris & Barto, 2009; Eysenbach et al., 2019; Sharma et al., 2020; Bagaria & Konidaris, 2019), and others adopting learned subgoal space (Vezhnevets et al., 2017; Li et al., 2021; 2023). Despite these variations, a common challenge is the difficulty of leveraging advancements in the field of off-policy, model-free RL.

Recent efforts to enhance HRL's learning efficiency through off-policy training have highlighted issues such as instability and the inherent non-stationarity problem of HRL. For instance, (Nachum et al., 2018) introduces an off-policy method that relabels past experiences to mitigate non-stationary effects on training. Techniques from hindsight experience replay have been utilized to train multi-level policies concurrently, penalizing high-level policies for unattainable subgoals (Andrychowicz et al., 2017a; Levy et al., 2019). To address the issue of large subgoal spaces, (Zhang et al., 2020) proposed constraining the high-level action space with an adjacency requirement. Wang et al.

(2020) improves stationarity by conditioning high-level decisions on both the low-level policy representation and environmental states. Li et al. (2021) develops a slowness objective for learning a subgoal representation function. Kim et al. (2021) introduces a framework for training a high-level policy with a reduced action space guided by landmarks, *i.e.*, promising states to explore. Adopting the deterministic subgoal representation of Li et al. (2021), Li et al. (2022) proposes an exploration strategy to enhance the high-level exploration via designing measures of novelty and potential for subgoals, albeit the strategy relies on on visit counts for subgoals in the constantly changing subgoal representation space.

The broader topic of goal generation in RL has also been explored (Florensa et al., 2018; Nair et al., 2018; Ren et al., 2019; Campero et al., 2021; Wang et al., 2023a). GoalGAN (Florensa et al., 2018) employs a GAN to generate appropriately challenging tasks for policy training, however it does not condition on observations and the sequential training of its GAN and policy. Nair et al. (2018) combines unsupervised representation learning with goal-conditioned policy training. Ren et al. (2019) proposes a method for generating immediately achievable hindsight goals. Campero et al. (2021) introduces a framework wherein a teacher network proposes progressively challenging goals, rewarding the network based on the student's performance. SAGA (Wang et al., 2023a) introduces an adversarially guided framework for generating subgoals in goal-conditioned HRL. However, akin to the common challenges associated with GANs, SAGA may encounter issues such as stability and mode collapse due to its implicit modeling of subgoal distributions. In contrast, HIDI explicitly constructs subgoal distributions by progressively transforming noise into sample data, effectively capturing multimodal distributions and ensuring more stable training dynamics.

Diffusion models have been introduced to offline RL domain recently (Janner et al., 2022; Wang et al., 2023b; Kang et al., 2024; Li et al., 2023; Chen et al., 2024). Janner et al. (2022) train an unconditional diffusion model to generate trajectories consisting of states and actions for offline RL. Approaches (Li et al., 2023; Chen et al., 2024) also extend diffusion model to offline HRL and generate trajectories at different levels. As a more related line of work, Diffsuion-QL (Wang et al., 2023b) introduces diffusion models into offline RL and demonstrated that diffusion models are superior at modeling complex action distributions. Kang et al. (2024) improve Diffsuion-QL to be compatible with maximum likelihood-based RL algorithms. The success of offline RL methods leveraging diffusion policies (Wang et al., 2023b; Kang et al., 2024) motivates us to investigate the impact of using conditional diffusion model for the challenging subgoal generation in off-policy HRL.

Gaussian processes can encode flexible priors over functions, which are a probabilistic machine learning paradigm (Williams & Rasmussen, 2006). GPs have been adopted in various latent variable modeling tasks in RL. In Engel et al. (2003), the use of GPs for solving the RL problem of value estimation is introduced. Then Kuss & Rasmussen (2003) uses GPs to model the the value function and system dynamics. Deisenroth et al. (2013) develops a GP-based transition model of a model-based learning system, which explicitly incorporates model uncertainty into long-term planning and controller learning to reduce the effects of model errors. Levine et al. (2011) proposes an algorithm for inverse reinforcement learning that represents nonlinear reward functions with GPs, allowing the recovery of both a reward function and the hyperparameters of a kernel function that describes the structure of the reward. Wang et al. (2024) proposes a GP based method for learning probabilistic subgoal representations in HRL.

## 4.1. Environments

Our experimental evaluation encompasses a diverse set of long-horizon continuous control tasks facilitated by the MuJoCo simulator (Todorov et al., 2012), which are widely adopted in the HRL community. The environments selected for testing our framework, depicted in Figure 3, include **Reacher**, **Pusher**, **Point Maze**, **Ant Maze (U-shape)**, **Ant Maze (W-shape)**, **Ant Fall**, **Ant FourRooms** and variants with environmental stochasticity for Ant Maze (U-shape), Ant Fall, and Ant FourRooms by adding Gaussian noise with a standard deviation of $\sigma = 0.05$ to the $(x, y)$ positions of the ant robot at each step. Additionally, we adopt another variant labeled 'Image' for the large-scale Ant FourRooms environment. In this variant, observations are low-resolution images formed by zeroing out the $(x, y)$ coordinates and appending a $5 \times 5 \times 3$ top-down view of the environment, as described in (Nachum et al., 2019; Li et al., 2021). We evaluate HIDI and baselines under dense and sparse reward paradigms. Dense rewards are the negative L2 distance to the target, while sparse rewards are 0 below a threshold and -1 otherwise. Maze tasks use a 2D $(x, y)$ goal space (Zhang et al., 2020; Kim et al., 2021), Reacher uses a 3D $(x, y, z)$ goal space, and Pusher employs a 6D space including the object's position. Relative subgoals are used in Maze tasks, and absolute schemes in Reacher and Pusher. [1]

## 4.2. Analysis

We conduct experiments comparing with the following state-of-the-art baseline methods: (1) **HLPS** (Wang et al., 2024): an HRL algorithm which proposes a GP-based subgoal latent space; (2) **SAGA** (Wang et al., 2023a): an HRL algo-

---

[1]Details about the environments and parameter configurations are provided in the appendix.

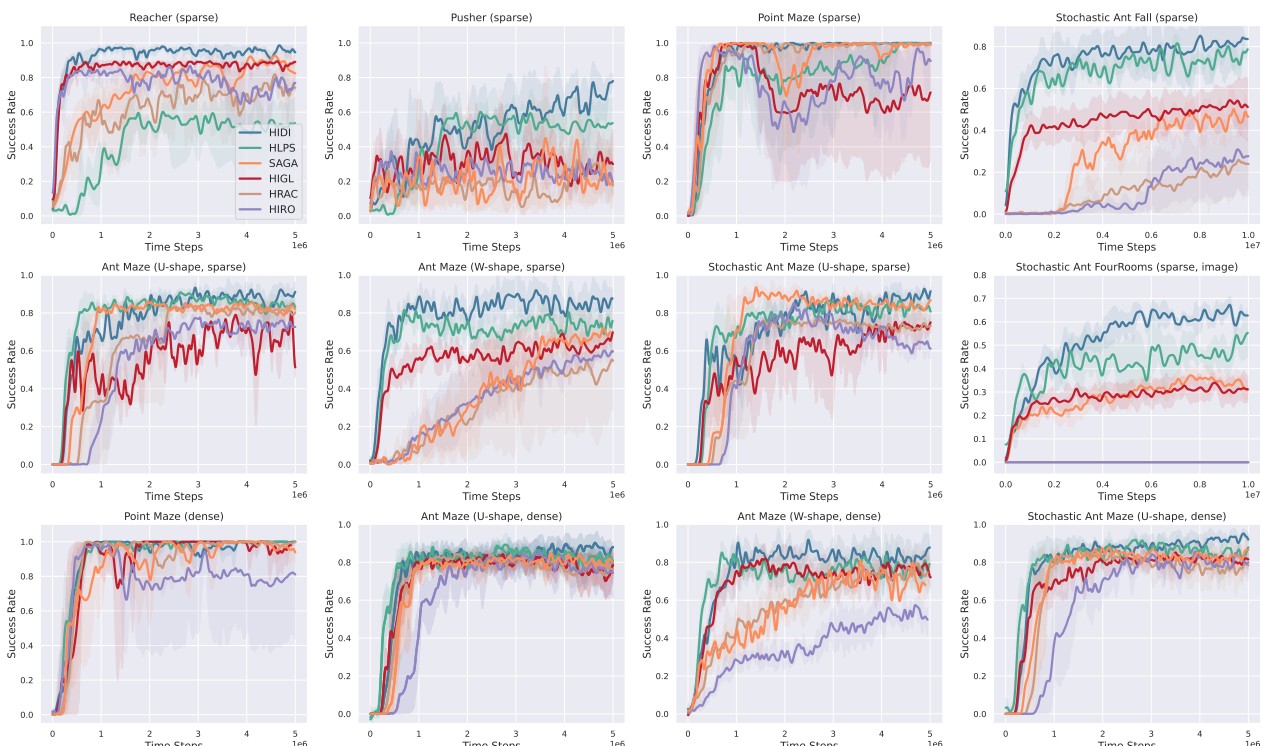

*Figure 1.* Learning curves of our method and baselines, *i.e.*, **HLPS** (Wang et al., 2024), **SAGA** (Wang et al., 2023a), **HIGL** (Kim et al., 2021), **HRAC** (Zhang et al., 2020), and **HIRO** (Nachum et al., 2018). Each curve and its shaded region represent the average success rate and 95% confidence interval respectively, averaged over 10 independent trials.

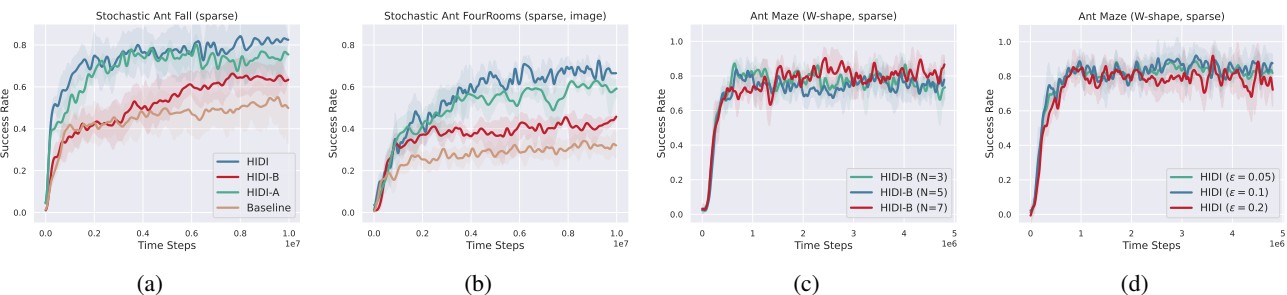

*Figure 2.* (a-b) Learning curves of various baselines: HIDI-A refers to HIDI without subgoal selection, HIDI-B refers to HIDI without subgoal selection and GP priors. (c) HIDI performance with varying diffusion steps. (d) HIDI performance with varying probabilities for subgoal selection.

rithm that introduces an adversarially guided framework for generating subgoals; (3) **HIGL** (Kim et al., 2021): an HRL algorithm that trains a high-level policy with a reduced subgoal space guided by landmarks; (4) **HRAC** (Zhang et al., 2020): an HRL algorithm which introduces an adjacency network to restrict the high-level action space to a $k$-step adjacent region of the current state; (5) **HIRO** (Nachum et al., 2018): an HRL algorithm that relabels the high-level actions based on hindsight experience (Andrychowicz et al., 2017b). Additionally, we present a theoretical analysis for HIDI in the Appendix A.2.

**Can HIDI surpass state-of-the-art HRL methods in learning stability and asymptotic performance?** Figure 1 illustrates the learning curves of HIDI in comparison with baseline methods across various tasks. HIDI consistently surpasses all baselines in terms of learning stability, sample efficiency, and asymptotic performance. The advantage of the hierarchical diffusion policy approach is more pronounced in complex scenarios such as the *Reacher* and *Pusher* robotic arm tasks, and the *Stochastic Ant Maze* task, where the environmental stochasticity poses additional challenges in generating reasonable subgoals. These results

highlight the benefits of employing generative models for subgoal generation, as demonstrated by HIDI and SAGA. However, SAGA exhibits signs of instability in the more demanding robotic arm task Pusher, as well as considerably lower sample efficiency compared to HIDI in most tasks.

**Is HIDI capable of generating reachable subgoals to address the issues commonly encountered in off-policy training within HRL?** Figure 5 in the Appendix illustrates generated subgoals and reached subgoals of HIDI and compared baselines in Ant Maze (W-shape, sparse) with the same starting location. This allows for an intuitive comparison of subgoals generated by HIDI, SAGA, HIGL, HRAC, and HIRO. Notably, HIDI generates reasonable subgoals for the lower level to achieve, as demonstrated by the small divergence between the generated and reached subgoals, which provides a stable learning signal for the low-level policy. In contrast, subgoals generated by HIRO are often unachievable and fail to guide the agent to reach the final target; subgoals generated by HRAC frequently get stuck in local minima due to its local adjacency constraint; HIGL and SAGA show improvement in both constraining the subgoals locally while jumping out of the local optimum of subgoals, yet the high-level policy is not adequately compatible with the low-level skills, *i.e.*, the increasing gap between the generated subgoal and reached subgoal leads to inferior performance compared with HIDI. This is further confirmed by the measure of distance between the generated subgoal and reached subgoal in Table 1. Subgoals generated by HLPS lie in a learned subgoal latent space and may not be quantitatively or qualitatively compared with other methods.

**How do various design choices within HIDI impact its empirical performance and effectiveness?** To understand the benefits of using a diffusion policy for generating subgoals, we constructed several baselines. *HIDI-A* denotes a baseline without performing the proposed subgoal selection strategy, *HIDI-B* is a baseline without adopting GP regularization and subgoal selection, while *Baseline*, adapted from a variant of the subgoal relabeling method *i.e.*, HIGL, is HIDI without the diffusion model, GP regularization, and subgoal selection strategy. Fig. 2 (a-b) illustrates the comparisons of various baselines and HIDI:

- Diffusional Subgoals: The performance improvement from *Baseline* to HIDI-B shows the benefit of adopting conditional diffusional model for subgoal generation, with a performance gain of ∼15%.

- Uncertainty Regularization: The sampling efficiency and performance improvement ∼15% and ∼16% respectively from HIDI-B to HIDI-A indicates the advantage of employing the GP prior on subgoal generation

as a surrogate distribution which potentially informs the diffusion process about uncertain areas. This highlights the GP's role in focusing learning on feasible regions consistent with past successful transitions.

- Subgoal Selection: Our proposed subgoal selection strategy demonstrates clear benefits. Comparing HIDI to HIDI-A shows performance improvements of approximately 7% and 8% on two challenging tasks. HIDI also achieves better sample efficiency compared to HIDI-A.

- Diffusion Steps $N$: Fig. 2 (c) shows the learning curves of baselines using varying number of diffusion steps $N$ used in HIDI. It empirically demonstrates that as $N$ increases from 3 to 7, the high-level policy becomes more expressive and capable of learning the more complex data distribution of subgoals. Since $N$ also serves as a trade-off between the expressiveness of subgoal modeling and computational complexity, we found that $N = 5$ is an efficient and effective setting for all the tasks during the experiment.

- Subgoal Selection Probability $\epsilon$: $\epsilon$ controls the chance to perform subgoal selection. As shown in Fig. 2 (d), when $\epsilon$ is large, *e.g.*, 0.25, the trade-off between the expressiveness of subgoal modeling and uncertainty information might be affected, *i.e.*, excessive subgoals sampled in uncertain regions may contribute to performance instability. When $\epsilon$ is small, *e.g.*, 0.05, the gain from subgoal selection would be decreased, and we set $\epsilon = 0.1$ for all other results.

- Scaling Factor $\eta$: We investigate the impact of $\eta$ in Eq. 4, which balances the diffusion objective and RL objective. As shown in Fig. 4 (Left), increasing $\eta$ improves performance at early training steps ($0 \sim 10^6$), while all three settings achieve similar performance at larger training steps ($4.2 \times 10^6 \sim 5 \times 10^6$). We report all other results based on $\eta = 5$ without loss of generality.

- Scaling Factor $\psi$: $\psi$ adjusts the influence of GP prior in learning the distribution of diffusional subgoals. As shown in Fig. 4 (Middle) in the appendix, increasing $\psi$ gives stronger GP prior and may slightly affect the flexibility of diffusion model learning, while decreasing $\psi$ renders the model to approximate baseline HIDI-B. We set $\psi = 10^{-3}$ for all other results.

## 5. Conclusion

In this paper, we presented a conditional diffusion model-based framework for subgoal generation in hierarchical reinforcement learning. By directly modeling a state-conditioned subgoal distribution, the approach mitigates

instabilities arising from off-policy training and leverages a Gaussian Process (GP) prior for explicit uncertainty quantification. We further proposed a subgoal selection strategy that integrates the diffusion model's expressiveness with the GP's structural guidance. This ensures subgoals align with meaningful patterns in the training data while maintaining robustness, supported by theoretical guarantees on regret and policy improvement. Experimental results across challenging continuous control benchmarks highlight the efficacy of this integrated approach in terms of both sample efficiency and performance.

## Acknowledgements

We acknowledge CSC – IT Center for Science, Finland, for awarding this project access to the LUMI supercomputer, owned by the EuroHPC Joint Undertaking, hosted by CSC (Finland) and the LUMI consortium through CSC. We acknowledge the computational resources provided by the Aalto Science-IT project. We acknowledge funding from Research Council of Finland (353198).

## Impact Statement

This paper presents work whose goal is to advance the field of Machine Learning. There are many potential societal consequences of our work, none which we feel must be specifically highlighted here.

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

# A. Appendix

## A.1. Theoretical Analysis of Diffusion-Based Subgoal Generation and GP Regularization

In this section, we provide a theoretical analysis for: (i) the effectiveness of the diffusion model for subgoal generation in hierarchical reinforcement learning (HRL), and (ii) the role of Gaussian Process (GP) regularization in guiding the diffusion process to generate reachable and structured subgoals.

### A.1.1. NOTATION AND PRELIMINARIES

Let:

- $s \in \mathcal{S}$ denote a state and $g \in \mathcal{G} \subseteq \mathcal{S}$ denote a subgoal.

- The high-level policy is parameterized by $\theta_h$ and defined as a reverse diffusion process:

$$\pi_{\theta_h}^h(g \mid s) = p_{\theta_h}(g_{0:N} \mid s) = \mathcal{N}(g_N; 0, I) \prod_{i=1}^{N} p_{\theta_h}(g_{i-1} \mid g_i, s), \tag{16}$$

  where $g_0 = g$ is the final generated subgoal and $N$ is the number of diffusion steps.

- The reverse diffusion step is defined as

$$g_{i-1} = \frac{1}{\sqrt{\alpha_i}} \left( g_i - \beta_i \frac{1}{\sqrt{1 - \bar{\alpha}_i}} \epsilon_{\theta_h}(g_i, s, i) \right) + \sqrt{\beta_i}\, \epsilon_i, \quad \epsilon_i \sim \mathcal{N}(0, I), \tag{17}$$

  with $\alpha_i = 1 - \beta_i$ and $\bar{\alpha}_i = \prod_{j=1}^{i} \alpha_j$.

- The diffusion model is trained using the loss

$$\mathcal{L}_{dm}(\theta_h) = \mathbb{E}_{i \sim \mathcal{U}(1,N),\, \epsilon \sim \mathcal{N}(0,I),\, (s,g) \sim D^h} \left[ \left\| \epsilon - \epsilon_{\theta_h}\left( \sqrt{\bar{\alpha}_i}\, g + \sqrt{1 - \bar{\alpha}_i}\, \epsilon,\, s,\, i \right) \right\|^2 \right], \tag{18}$$

  where $D^h$ is the high-level replay buffer of relabeled subgoals.

- A Gaussian Process (GP) prior is placed on the subgoal distribution conditioned on $s$:

$$p(g \mid s; \theta_{gp}) = \mathcal{N}\big(g; 0, K_N + \sigma^2 I\big), \tag{19}$$

  where $K_N$ is the covariance matrix defined via a kernel function $K(s, s')$, and $\sigma^2$ is the noise variance.

- The GP regularization loss is defined as:

$$\mathcal{L}_{gp}(\theta_h, \theta_{gp}) = \mathbb{E}_{(s,g) \sim D^h} \left[ -\log p(g \mid s; \theta_{gp}) \right]. \tag{20}$$

- The overall high-level training objective combines the diffusion loss, GP regularization (weighted by $\psi$), and an RL objective $\mathcal{L}_{dpg}(\theta_h)$:

$$L(\theta_h) = \mathcal{L}_{dm}(\theta_h) + \psi\, \mathcal{L}_{gp}(\theta_h, \theta_{gp}) + \eta\, \mathcal{L}_{dpg}(\theta_h). \tag{21}$$

### A.1.2. VALIDITY OF THE LEARNED SUBGOAL DISTRIBUTION VIA DIFFUSION

We first establish conditions under which the reverse diffusion process yields a subgoal distribution close to the target distribution $q(g \mid s)$ (defined by the relabeled data in $D^h$).

**Theorem A.1** (Validity of the Learned Subgoal Distribution). *Let $q(g|s)$ be the target distribution of subgoals given state $s$ (implicitly defined by the relabeled data in $\mathcal{D}_h$). Let $p_{\theta_h}(g|s)$ be the distribution generated by the reverse diffusion process parameterized by $\theta_h$. Assume the noise prediction network $\epsilon_{\theta_h}$ is trained by minimizing the simplified objective:*

$$\mathcal{L}_{dm}(\theta_h) = \mathbb{E}_{i \sim \mathcal{U}(1,N),\, \epsilon \sim \mathcal{N}(0,I),\, (s,g_0) \sim \mathcal{D}^h} \left[ \left\| \epsilon - \epsilon_{\theta_h}\left( \sqrt{\bar{\alpha}_i}\, g_0 + \sqrt{1 - \bar{\alpha}_i}\, \epsilon,\, s,\, i \right) \right\|^2 \right]. \tag{22}$$

*If the expected loss achieved by the optimal parameters $\theta_h^*$ is bounded, $\mathbb{E}[\mathcal{L}_{dm}(\theta_h^*)] \leq \delta^2$, then the KL divergence between the target distribution and the learned distribution is bounded, $D_{KL}(q(g|s)||p_{\theta_h^*}(g|s)) \leq B(\delta, N, \{\beta_i\})$, where $B$ is a function that depends on the noise prediction error $\delta$, the number of steps $N$, and the noise schedule $\{\beta_i\}$, satisfying $\lim_{\delta \to 0} B(\delta, \dots) = 0$. Consequently, by Pinsker's inequality, the total variation distance is also bounded:*

$$\|q(g|s) - p_{\theta_h^*}(g|s)\|_{TV} \leq \sqrt{\frac{1}{2}D_{KL}(q(g|s)||p_{\theta_h^*}(g|s))} \leq \sqrt{\frac{1}{2}B(\delta, N, \{\beta_i\})}. \tag{23}$$

*Proof.* The proof relies on connecting the simplified objective $\mathcal{L}_{dm}$ to the variational lower bound (ELBO) on the log-likelihood $\log p_{\theta_h}(g_0|s)$. The ELBO for the reverse process $p_{\theta_h}(g_0|s)$ conditioned on $s$, marginalizing over $g_{1:N}$, is:

$$\log p_{\theta_h}(g_0|s) \geq \mathbb{E}_{q(g_{1:N}|g_0,s)}\left[\log \frac{p_{\theta_h}(g_{0:N}|s)}{q(g_{1:N}|g_0,s)}\right] \tag{24}$$

$$= \mathbb{E}_q\left[\log p(g_N) + \sum_{i=1}^N \log p_{\theta_h}(g_{i-1}|g_i, s) - \sum_{i=1}^N \log q(g_i|g_{i-1}, s)\right] \tag{25}$$

$$= \underbrace{\mathbb{E}_q[\log p_{\theta_h}(g_0|g_1, s)]}_{\text{Reconstruction Term}} - \underbrace{\mathbb{E}_q\left[\sum_{i=2}^N D_{KL}(q(g_{i-1}|g_i, g_0, s)||p_{\theta_h}(g_{i-1}|g_i, s))\right]}_{\text{KL terms}}$$

$$- \underbrace{D_{KL}(q(g_N|g_0, s)||p(g_N))}_{\text{Prior Matching Term}} \tag{26}$$

where $q(g_{i-1}|g_i, g_0, s)$ is the true posterior of the forward process, which is tractable and Gaussian: $q(g_{i-1}|g_i, g_0, s) = \mathcal{N}(g_{i-1}; \tilde{\mu}_i(g_i, g_0), \tilde{\beta}_i I)$ with $\tilde{\mu}_i(g_i, g_0) = \frac{\sqrt{\bar{\alpha}_{i-1}}\beta_i}{1-\bar{\alpha}_i}g_0 + \frac{\sqrt{\alpha_i}(1-\bar{\alpha}_{i-1})}{1-\bar{\alpha}_i}g_i$ and $\tilde{\beta}_i = \frac{1-\bar{\alpha}_{i-1}}{1-\bar{\alpha}_i}\beta_i$.

The diffusion model $p_{\theta_h}(g_{i-1}|g_i, s) = \mathcal{N}(g_{i-1}; \mu_{\theta_h}(g_i, s, i), \beta_i I)$ uses a learned mean $\mu_{\theta_h}(g_i, s, i) = \frac{1}{\sqrt{\alpha_i}}\left(g_i - \frac{\beta_i}{\sqrt{1-\bar{\alpha}_i}}\epsilon_{\theta_h}(g_i, s, i)\right)$ and a fixed variance $\beta_i I$ (or sometimes $\tilde{\beta}_i I$).

Ho et al. (2020) shows that minimizing the simplified objective $\mathcal{L}_{dm}(\theta_h)$ (Eq. 22) is equivalent to optimizing a weighted sum of the KL terms in the ELBO (26), specifically minimizing the discrepancy between the predicted noise $\epsilon_{\theta_h}$ and the actual noise $\epsilon$ used to generate $g_i = \sqrt{\bar{\alpha}_i}g_0 + \sqrt{1-\bar{\alpha}_i}\epsilon$. Let $\mathcal{L}_{VLB}(\theta_h) = -\mathbb{E}_{q(g_0|s)}[\text{ELBO for } g_0]$. Then:

$$D_{KL}(q(g_0|s)||p_{\theta_h}(g_0|s)) = \mathcal{L}_{VLB}(\theta_h) - H(q(g_0|s)), \tag{27}$$

where $H(q(g_0|s))$ is the entropy of the true data distribution, which is constant w.r.t. $\theta_h$. Minimizing $\mathcal{L}_{dm}$ upper bounds the terms contributing to $\mathcal{L}_{VLB}$. Specifically, the expected squared error in noise prediction relates to the KL divergence terms:

$$\mathbb{E}_{g_0,\epsilon}[\|\epsilon - \epsilon_{\theta_h}(\sqrt{\bar{\alpha}_i}g_0 + \sqrt{1-\bar{\alpha}_i}\epsilon, s, i)\|^2] = C_i \mathbb{E}_{g_0}[D_{KL}(q(g_{i-1}|g_i, g_0, s)||p_{\theta_h}(g_{i-1}|g_i, s))] + \text{const}, \tag{28}$$

where $C_i = \frac{2\sigma_i^2(1-\alpha_i)}{\beta_i^2}$ depends on the variance $\sigma_i^2 = \beta_i$ or $\tilde{\beta}_i$ used in $p_{\theta_h}$. (See Appendix B of (Ho et al., 2020) for details).

Summing over $i$ (with appropriate weighting, often simplified in practice as in $\mathcal{L}_{dm}$), if $\mathbb{E}[\mathcal{L}_{dm}(\theta_h^*)] \leq \delta^2$, it implies that the sum of the KL divergence terms in the ELBO expression (26) is bounded by some function $B'(\delta, N, \{\beta_i\})$.

$$\mathcal{L}_{VLB}(\theta_h^*) \leq \mathbb{E}_{q(g_0|s)}[-\log p_{\theta_h^*}(g_0|g_1, s)] + B'(\delta, N, \{\beta_i\}) + D_{KL}(q(g_N|g_0, s)||p(g_N)) \tag{29}$$

If we also assume the reconstruction term is well-behaved (or absorbed into the bound) and the prior matching term is small (as $T \to \infty$), minimizing $\mathcal{L}_{dm}$ effectively minimizes an upper bound on $\mathcal{L}_{VLB}$. Thus, $D_{KL}(q(g_0|s)||p_{\theta_h^*}(g_0|s)) = \mathcal{L}_{VLB}(\theta_h^*) - H(q(g_0|s))$ is bounded by a function $B(\delta, N, \{\beta_i\})$ which incorporates $B'$ and other terms, and vanishes as $\delta \to 0$.

Finally, applying Pinsker's inequality, $\|q(g|s) - p_{\theta_h^*}(g|s)\|_{TV}^2 \leq \frac{1}{2}D_{KL}(q(g|s)||p_{\theta_h^*}(g|s))$, yields the desired bound on the total variation distance. $\square$

A.1.3. GP REGULARIZATION AND ITS INTERACTION WITH THE DIFFUSION MODEL

We now analyze how the GP regularization term influences the generated subgoal and guides the diffusion process.

**Theorem A.2** (Guiding Effect of GP Regularization). *Let the overall loss function for the high-level policy parameters $\theta_h$ be*

$$L(\theta_h) = \mathcal{L}_{dm}(\theta_h) + \psi \, \mathcal{L}_{gp}(\theta_h, \theta_{gp}) + \eta \, \mathcal{L}_{dpg}(\theta_h).$$

*The GP regularization term is defined using the reparameterization trick, where*

$$\mathbf{g} = f(\boldsymbol{\epsilon}', \mathbf{s}; \theta_h)$$

*is the subgoal generated by the reverse diffusion process from base noise $\boldsymbol{\epsilon}' \sim \mathcal{N}(0, I)$ and state condition $\mathbf{s}$:*

$$\mathcal{L}_{gp}(\theta_h, \theta_{gp}) = \mathbb{E}_{\mathbf{s} \sim \mathcal{D}_h, \, \boldsymbol{\epsilon}' \sim \mathcal{N}(0,I)} \left[ -\log p_{GP}\Big( f(\boldsymbol{\epsilon}', \mathbf{s}; \theta_h) \mid \mathbf{s}; \mathcal{D}_h, \theta_{gp} \Big) \right], \tag{30}$$

*where*

$$p_{GP}(\mathbf{g} \mid \mathbf{s}; \mathcal{D}_h, \theta_{gp}) = \mathcal{N}(\mathbf{g} \mid \mu_*(\mathbf{s}), \sigma_*^2(\mathbf{s})I)$$

*is the predictive distribution of the (sparse) GP conditioned on the high-level data $\mathcal{D}_h$ (see Eq. 39), assuming isotropic variance for simplicity. The gradient of this term with respect to the diffusion model parameters $\theta_h$ is given by:*

$$\nabla_{\theta_h} \mathcal{L}_{gp} = \mathbb{E}_{\mathbf{s} \sim \mathcal{D}_h, \, \boldsymbol{\epsilon}' \sim \mathcal{N}(0,I)} \left[ \left( \frac{\mathbf{g} - \mu_*(\mathbf{s})}{\sigma_*^2(\mathbf{s})} \right)^\top \nabla_{\theta_h} \mathbf{g} \right], \tag{31}$$

*where $\mathbf{g} = f(\boldsymbol{\epsilon}', \mathbf{s}; \theta_h)$. This gradient term encourages the parameters $\theta_h$ to be updated during training such that the generated subgoals $\mathbf{g}$ tend to move closer to the GP predictive mean $\mu_*(\mathbf{s})$, with the influence being stronger in regions where the GP has low predictive variance $\sigma_*^2(\mathbf{s})$.*

*Proof.* We analyze the gradient of the GP loss term $\mathcal{L}_{gp}$ with respect to the parameters $\theta_h$ of the diffusion model, which defines the reverse process $p_{\theta_h}(\mathbf{g} \mid \mathbf{s})$.

The subgoal $\mathbf{g}$ generated by the diffusion model is given by

$$\mathbf{g} = f(\boldsymbol{\epsilon}', \mathbf{s}; \theta_h),$$

where $\boldsymbol{\epsilon}' \sim \mathcal{N}(0, I)$ and $f(\cdot)$ is differentiable with respect to $\theta_h$. Using the reparameterization trick, we write:

$$
\begin{aligned}
\nabla_{\theta_h} \mathcal{L}_{gp} &= \nabla_{\theta_h} \mathbb{E}_{\mathbf{s} \sim \mathcal{D}_h, \, \boldsymbol{\epsilon}' \sim \mathcal{N}(0,I)} \left[ -\log p_{GP}\Big( f(\boldsymbol{\epsilon}', \mathbf{s}; \theta_h) \mid \mathbf{s}; \mathcal{D}_h, \theta_{gp} \Big) \right] \\
&= \mathbb{E}_{\mathbf{s} \sim \mathcal{D}_h, \, \boldsymbol{\epsilon}' \sim \mathcal{N}(0,I)} \left[ \nabla_{\theta_h} \Big( -\log p_{GP}\big( f(\boldsymbol{\epsilon}', \mathbf{s}; \theta_h) \mid \mathbf{s}; \mathcal{D}_h, \theta_{gp} \big) \Big) \right] \\
&= \mathbb{E}_{\mathbf{s} \sim \mathcal{D}_h, \, \boldsymbol{\epsilon}' \sim \mathcal{N}(0,I)} \left[ \left( \nabla_{\mathbf{g}} \left( -\log p_{GP}(\mathbf{g} \mid \mathbf{s}; \mathcal{D}_h, \theta_{gp}) \right) \right)^\top \nabla_{\theta_h} \mathbf{g} \right]_{\mathbf{g} = f(\boldsymbol{\epsilon}', \mathbf{s}; \theta_h)}.
\end{aligned}
$$

For the Gaussian predictive distribution

$$p_{GP}(\mathbf{g} \mid \mathbf{s}) = \mathcal{N}(\mathbf{g} \mid \mu_*(\mathbf{s}), \sigma_*^2(\mathbf{s})I),$$

the negative log-likelihood is

$$-\log p_{GP}(\mathbf{g} \mid \mathbf{s}) = \frac{1}{2\sigma_*^2(\mathbf{s})} \|\mathbf{g} - \mu_*(\mathbf{s})\|^2 + \frac{D}{2} \log(2\pi\sigma_*^2(\mathbf{s})),$$

where $D$ is the dimensionality of $\mathbf{g}$. Differentiating with respect to $\mathbf{g}$ yields

$$\nabla_{\mathbf{g}} \left[ -\log p_{GP}(\mathbf{g} \mid \mathbf{s}) \right] = \frac{\mathbf{g} - \mu_*(\mathbf{s})}{\sigma_*^2(\mathbf{s})}.$$

Substituting this into our expression, we obtain

$$\nabla_{\theta_h}\mathcal{L}_{gp} = \mathbb{E}_{\mathbf{s}\sim\mathcal{D}_h,\,\boldsymbol{\epsilon}'\sim\mathcal{N}(0,I)}\left[\left(\frac{\mathbf{g}-\mu_*(\mathbf{s})}{\sigma_*^2(\mathbf{s})}\right)^{\top}\nabla_{\theta_h}\mathbf{g}\right].$$

During optimization via gradient descent, the update for $\theta_h$ is given by

$$\theta_h \leftarrow \theta_h - \alpha\nabla_{\theta_h}L(\theta_h),$$

with $\alpha$ being the learning rate. The contribution from the GP loss term is $-\alpha\psi\nabla_{\theta_h}\mathcal{L}_{gp}$, which effectively drives the parameters $\theta_h$ to adjust so that the generated subgoal $\mathbf{g}$ aligns more closely with the GP predictive mean $\mu_*(\mathbf{s})$, particularly when $\sigma_*^2(\mathbf{s})$ is small (indicating high confidence).

This completes the proof. □

*Remark* A.3 (Connection to KL Divergence). The GP regularization term $\mathcal{L}_{gp}$ can be related to the KL divergence between the learned conditional distribution $p_{\theta_h}(\mathbf{g}\mid\mathbf{s})$ and the GP predictive distribution

$$p_{GP}(\mathbf{g}\mid\mathbf{s}) = \mathcal{N}(\mathbf{g}\mid\mu_*(\mathbf{s}),\sigma_*^2(\mathbf{s})I).$$

Specifically,

$$\mathcal{L}_{gp}(\theta_h,\theta_{gp}) = \mathbb{E}_{\mathbf{s}\sim\mathcal{D}_h}\left[\mathbb{E}_{\mathbf{g}\sim p_{\theta_h}(\cdot\mid\mathbf{s})}\left[-\log p_{GP}(\mathbf{g}\mid\mathbf{s})\right]\right]$$
$$= \mathbb{E}_{\mathbf{s}\sim\mathcal{D}_h}\left[D_{KL}\Big(p_{\theta_h}(\cdot\mid\mathbf{s})\Big\|p_{GP}(\cdot\mid\mathbf{s})\Big) + H\Big(p_{\theta_h}(\cdot\mid\mathbf{s})\Big)\right],$$

where $H(\cdot)$ denotes differential entropy. Thus, minimizing $\mathcal{L}_{gp}$ reduces the KL divergence between the learned subgoal distribution and the GP predictive distribution, thereby encouraging the diffusion model to produce subgoals that are aligned with the GP's predictions.

### A.1.4. DISCUSSION

**Diffusion Model Validity.** Theorem A.1 demonstrates that if the noise prediction error is sufficiently small, the reverse diffusion process accurately approximates the target state-conditioned subgoal distribution. Hence, under reasonable conditions, the diffusion model is theoretically justified for subgoal generation in HRL.

**GP Regularization Impact.** Theorem A.2 shows that the GP regularization term actively pulls the generated subgoal towards the GP predictive mean. In regions where the GP is confident (i.e., $\lambda_{\min}(K_N+\sigma^2 I)$ is large), the generated subgoal is tightly coupled with the observed, reachable subgoals. This guides the diffusion model, ensuring that even if the learned distribution is inherently non-stationary, the GP regularization mitigates extreme deviations by serving as a principled, uncertainty-aware anchor.

### A.2. Theoretical Analysis of Subgoal Selection

**Assumption A.4** (Near-Optimal Diffusion Policy (for Single-Step)). We assume that after sufficient training and data coverage, for every $\mathbf{s}\in\mathcal{S}$,

$$\mathbb{E}_{\mathbf{g}\sim\pi_{\theta_h}(\cdot\mid\mathbf{s})}[R(\mathbf{s},\mathbf{g})] \geq \max_{\mathbf{g}'\in\mathcal{G}}R(\mathbf{s},\mathbf{g}') - \delta,$$

for some small $\delta > 0$. This states that the *diffusion-based* subgoal distribution $\pi_{\theta_h}$ nearly achieves the maximum possible single-step reward at each $\mathbf{s}$.

*Detailed Proof of Theorem 3.3.* Let $R^*(\mathbf{s}) = \max_{\mathbf{g}\in\mathcal{G}}R(\mathbf{s},\mathbf{g})$ be the optimal single-step reward. The regret of the subgoal selection strategy $\widetilde{\pi}_h$ at state $\mathbf{s}$ is given by $R^*(\mathbf{s}) - \mathbb{E}_{\mathbf{g}\sim\widetilde{\pi}_h}[R(\mathbf{s},\mathbf{g})]$.

By the definition of the subgoal selection strategy $\widetilde{\pi}_h(\mathbf{g}|\mathbf{s}) = \varepsilon\, \delta_{\boldsymbol{\mu}(\mathbf{s})}(\mathbf{g}) + (1-\varepsilon)\, \pi_{\theta_h}(\mathbf{g}|\mathbf{s})$, the expected reward under this policy is:

$$\mathbb{E}_{\mathbf{g}\sim\widetilde{\pi}_h}[R(\mathbf{s},\mathbf{g})] = \int_{\mathcal{G}} R(\mathbf{s},\mathbf{g})\widetilde{\pi}_h(\mathbf{g}|\mathbf{s})d\mathbf{g}$$

$$= \int_{\mathcal{G}} R(\mathbf{s},\mathbf{g})\left[\varepsilon\, \delta_{\boldsymbol{\mu}(\mathbf{s})}(\mathbf{g}) + (1-\varepsilon)\, \pi_{\theta_h}(\mathbf{g}|\mathbf{s})\right]d\mathbf{g}$$

$$= \varepsilon \int_{\mathcal{G}} R(\mathbf{s},\mathbf{g})\delta_{\boldsymbol{\mu}(\mathbf{s})}(\mathbf{g})d\mathbf{g} + (1-\varepsilon)\int_{\mathcal{G}} R(\mathbf{s},\mathbf{g})\pi_{\theta_h}(\mathbf{g}|\mathbf{s})d\mathbf{g}$$

$$= \varepsilon\, R\big(\mathbf{s},\boldsymbol{\mu}(\mathbf{s})\big) + (1-\varepsilon)\, \mathbb{E}_{\mathbf{g}\sim\pi_{\theta_h}}[R(\mathbf{s},\mathbf{g})],$$

where we used the property of the Dirac delta function in the last step.

Given that $R\big(\mathbf{s},\boldsymbol{\mu}(\mathbf{s})\big) \geq R_{\min}$ by assumption, and under Assumption A.4, we have $\mathbb{E}_{\mathbf{g}\sim\pi_{\theta_h}}[R(\mathbf{s},\mathbf{g})] \geq R^*(\mathbf{s}) - \delta$, we can lower bound the expected reward:

$$\mathbb{E}_{\mathbf{g}\sim\widetilde{\pi}_h}[R(\mathbf{s},\mathbf{g})] \geq \varepsilon\, R_{\min} + (1-\varepsilon)\, (R^*(\mathbf{s}) - \delta).$$

Now, we can bound the single-step regret:

$$R^*(\mathbf{s}) - \mathbb{E}_{\mathbf{g}\sim\widetilde{\pi}_h}[R(\mathbf{s},\mathbf{g})] \leq R^*(\mathbf{s}) - \left[\varepsilon\, R_{\min} + (1-\varepsilon)\, (R^*(\mathbf{s}) - \delta)\right]$$

$$= R^*(\mathbf{s}) - \varepsilon R_{\min} - (1-\varepsilon)R^*(\mathbf{s}) + (1-\varepsilon)\delta$$

$$= \varepsilon R^*(\mathbf{s}) - \varepsilon R_{\min} + (1-\varepsilon)\delta$$

$$= \varepsilon(R^*(\mathbf{s}) - R_{\min}) + (1-\varepsilon)\delta.$$

Thus, the single-step regret of the subgoal selection strategy is bounded by $\varepsilon\left(R^*(\mathbf{s}) - R_{\min}\right) + (1-\varepsilon)\,\delta$. $\qquad\square$

*Detailed Proof of Proposition 3.4.* This proof relies on the principle of policy improvement, a fundamental concept in reinforcement learning (Sutton & Barto, 2018). The value function of a policy $\pi$ is given by $J(\pi) = \mathbb{E}_{\mathbf{s}_0\sim d_0, \mathbf{g}_t\sim\pi(\cdot|\mathbf{s}_t)}\left[\sum_{t=0}^{\infty}\gamma^t r_t^h\right]$, where $r_t^h$ is the high-level reward. A single step of policy improvement involves changing the policy in a way that increases the value function.

Consider the expected Q-value under the subgoal selection strategy $\widetilde{\pi}_h$ at state $\mathbf{s}$:

$$\mathbb{E}_{\mathbf{g}\sim\widetilde{\pi}_h(\cdot|\mathbf{s})}[Q_h(\mathbf{s},\mathbf{g})] = \int_{\mathcal{G}} Q_h(\mathbf{s},\mathbf{g})\widetilde{\pi}_h(\mathbf{g}|\mathbf{s})d\mathbf{g}$$

$$= \int_{\mathcal{G}} Q_h(\mathbf{s},\mathbf{g})\left[\varepsilon\, \delta_{\boldsymbol{\mu}(\mathbf{s})}(\mathbf{g}) + (1-\varepsilon)\, \pi_{\theta_h}(\mathbf{g}|\mathbf{s})\right]d\mathbf{g}$$

$$= \varepsilon\, Q_h\big(\mathbf{s},\boldsymbol{\mu}(\mathbf{s})\big) + (1-\varepsilon)\, \mathbb{E}_{\mathbf{g}\sim\pi_{\theta_h}(\cdot|\mathbf{s})}[Q_h(\mathbf{s},\mathbf{g})].$$

Assume the high-level Q-function $Q_h(\mathbf{s},\mathbf{g})$ is *Lipschitz smooth* in $\mathbf{g}$ with constant $L$:

$$|Q_h(\mathbf{s},\mathbf{g}_1) - Q_h(\mathbf{s},\mathbf{g}_2)| \leq L\|\mathbf{g}_1 - \mathbf{g}_2\|, \quad \forall \mathbf{g}_1, \mathbf{g}_2 \in \mathcal{G}. \tag{32}$$

If the GP's predictive mean $\boldsymbol{\mu}(\mathbf{s})$ is close to subgoals $\mathbf{g}$ with high $Q_h$, then $\boldsymbol{\mu}(\mathbf{s})$ will inherit high Q-values through smoothness. The GP is trained on state-subgoal pairs $(\mathbf{s},\mathbf{g})$ from the replay buffer $\mathcal{B}_h$, containing high-reward transitions (by definition of $R(\mathbf{s},\mathbf{g})$). The GP kernel $k(\cdot,\cdot)$ captures correlations in $\mathcal{S}\times\mathcal{G}$, enabling generalization of high-Q subgoals to new states.

The predictive mean $\boldsymbol{\mu}(\mathbf{s})$ minimizes the posterior expected squared error:

$$\boldsymbol{\mu}(\mathbf{s}) = \arg\min_{\mathbf{g}'} \mathbb{E}_{\mathbf{g}\sim p(\mathbf{g}|\mathbf{s},\mathcal{B}_h)}\left[\|\mathbf{g}' - \mathbf{g}\|^2\right], \tag{33}$$

where $p(\mathbf{g}|\mathbf{s},\mathcal{B}_h)$ is the posterior subgoal distribution. If high-Q subgoals in $\mathcal{B}_h$ cluster around $\boldsymbol{\mu}(\mathbf{s})$, then:

$$Q_h(\mathbf{s},\boldsymbol{\mu}(\mathbf{s})) \geq \mathbb{E}_{\mathbf{g}\sim p(\mathbf{g}|\mathbf{s},\mathcal{B}_h)}[Q_h(\mathbf{s},\mathbf{g})]. \tag{34}$$

This holds under *unimodality* or *concentration* of high-Q subgoals in $\mathcal{B}_h$.

The diffusion policy $\pi_{\theta_h}$ generates subgoals through a stochastic generative process. TD3 employs deterministic high-level actions with exploration noise. However, the diffusion model's sampling process inherently encourages diversity by gradually denoising from a Gaussian distribution. This results in subgoals distributed around high-reward regions in $\mathcal{G}$, with a trade-off between exploitation (high-$Q_h$ subgoals) and exploration (diverse subgoals).

Let $\mathbf{g}^* = \arg\max_{\mathbf{g}} Q_h(\mathbf{s}, \mathbf{g})$ denote the optimal subgoal. The diffusion policy $\pi_{\theta_h}$ samples subgoals such that:

$$\mathbb{E}_{\mathbf{g} \sim \pi_{\theta_h}}[Q_h(\mathbf{s}, \mathbf{g})] \leq Q_h(\mathbf{s}, \mathbf{g}^*) - \Delta,$$

where $\Delta > 0$ quantifies the exploration penalty due to the diffusion process's stochasticity. Meanwhile, the GP mean $\boldsymbol{\mu}(\mathbf{s})$ acts as a *deterministic proxy* for subgoals frequently visited in high-reward trajectories within $\mathcal{B}_h$. If $\boldsymbol{\mu}(\mathbf{s})$ approximates $\mathbf{g}^*$ with error $\epsilon$ (i.e., $\|\boldsymbol{\mu}(\mathbf{s}) - \mathbf{g}^*\| \leq \epsilon$), Lipschitz continuity of $Q_h$ implies:

$$Q_h(\mathbf{s}, \boldsymbol{\mu}(\mathbf{s})) \geq Q_h(\mathbf{s}, \mathbf{g}^*) - L\epsilon.$$

Thus, for $L\epsilon < \Delta$, we have:

$$Q_h(\mathbf{s}, \boldsymbol{\mu}(\mathbf{s})) \geq Q_h(\mathbf{s}, \mathbf{g}^*) - L\epsilon \geq \mathbb{E}_{\mathbf{g} \sim \pi_{\theta_h}}[Q_h(\mathbf{s}, \mathbf{g})].$$

Therefore:

$$\mathbb{E}_{\mathbf{g} \sim \widetilde{\pi}_h(\cdot|\mathbf{s})}[Q_h(\mathbf{s}, \mathbf{g})] \geq \varepsilon\, \mathbb{E}_{\mathbf{g} \sim \pi_{\theta_h}(\cdot|\mathbf{s})}[Q_h(\mathbf{s}, \mathbf{g})] + (1 - \varepsilon)\, \mathbb{E}_{\mathbf{g} \sim \pi_{\theta_h}(\cdot|\mathbf{s})}[Q_h(\mathbf{s}, \mathbf{g})]$$
$$= \mathbb{E}_{\mathbf{g} \sim \pi_{\theta_h}(\cdot|\mathbf{s})}[Q_h(\mathbf{s}, \mathbf{g})].$$

The value function can be expressed as $J(\pi) = \mathbb{E}_{\mathbf{s} \sim d_\pi}\left[\mathbb{E}_{\mathbf{g} \sim \pi(\cdot|\mathbf{s})}[Q_h(\mathbf{s}, \mathbf{g})]\right]$, where $d_\pi$ is the stationary distribution of states under policy $\pi$.

If the induced state distribution $d_{\widetilde{\pi}_h}$ remains close to $d_{\pi_{\theta_h}}$, which is a common assumption for a single-step policy improvement analysis as the change in policy is controlled by $\varepsilon$, then:

$$J(\widetilde{\pi}_h) = \mathbb{E}_{\mathbf{s} \sim d_{\widetilde{\pi}_h}}[\mathbb{E}_{\mathbf{g} \sim \widetilde{\pi}_h}[Q_h(\mathbf{s}, \mathbf{g})]]$$
$$\geq \mathbb{E}_{\mathbf{s} \sim d_{\widetilde{\pi}_h}}\left[\mathbb{E}_{\mathbf{g} \sim \pi_{\theta_h}}[Q_h(\mathbf{s}, \mathbf{g})]\right]$$
$$\approx \mathbb{E}_{\mathbf{s} \sim d_{\pi_{\theta_h}}}\left[\mathbb{E}_{\mathbf{g} \sim \pi_{\theta_h}}[Q_h(\mathbf{s}, \mathbf{g})]\right]$$
$$= J(\pi_{\theta_h}).$$

Thus, $J(\widetilde{\pi}_h) \geq J(\pi_{\theta_h})$, indicating that the subgoal selection strategy achieves a value function at least as good as the original policy $\pi_{\theta_h}$ in a single step. $\qquad\square$

### A.3. Sparse Gaussian Process Derivation

As mentioned in the main text, standard Gaussian Process (GP) regression involves inverting an $N \times N$ covariance matrix, where $N$ is the number of data points in the high-level replay buffer $\mathcal{D}_h$. This has a computational complexity of $\mathcal{O}(N^3)$, which is prohibitive for large datasets typical in RL. To address this, we employ a sparse GP approximation using inducing points (Snelson & Ghahramani, 2005).

We introduce a smaller set of $M$ inducing states $\bar{\mathbf{s}} = \{\bar{\mathbf{s}}_m\}_{m=1}^M$, where $M \ll N$. These inducing states are associated with corresponding "imaginary" target subgoals $\bar{\mathbf{g}} = \{\bar{\mathbf{g}}_m\}_{m=1}^M$. The core idea is that the predictive distribution for any new state $\mathbf{s}_*$ depends only on these $M$ inducing points, which summarize the full dataset.

Let $\bar{\mathcal{D}} = (\bar{\mathbf{s}}, \bar{\mathbf{g}})$ denote the pseudo-dataset of inducing points. The joint distribution of the observed subgoals $\mathbf{g}$ (from $\mathcal{D}_h$) and the imaginary subgoals $\bar{\mathbf{g}}$ under the GP prior is:

$$p(\mathbf{g}, \bar{\mathbf{g}} | \mathbf{s}, \bar{\mathbf{s}}) = \mathcal{N}\left(\begin{bmatrix} \mathbf{g} \\ \bar{\mathbf{g}} \end{bmatrix} \middle| \mathbf{0}, \begin{bmatrix} \mathbf{K}_{NN} + \sigma^2 \mathbf{I} & \mathbf{K}_{NM} \\ \mathbf{K}_{MN} & \mathbf{K}_{MM} \end{bmatrix}\right),$$

where $\mathbf{K}_{NN}$, $\mathbf{K}_{NM}$, $\mathbf{K}_{MN} = \mathbf{K}_{NM}^\top$, and $\mathbf{K}_{MM} = \mathbf{K}_M$ are covariance matrices computed using the kernel function $K$ between the states $\mathbf{s}$ in $\mathcal{D}_h$ and the inducing states $\bar{\mathbf{s}}$. Specifically, $[\mathbf{K}_{NM}]_{nm} = K(\mathbf{s}_n, \bar{\mathbf{s}}_m)$ and $[\mathbf{K}_{MM}]_{mm'} = K(\bar{\mathbf{s}}_m, \bar{\mathbf{s}}_{m'})$.

The sparse approximation assumes that the observed subgoals $\mathbf{g}$ are conditionally independent given the imaginary subgoals $\bar{\mathbf{g}}$. The likelihood of a single subgoal $\mathbf{g}$ given a state $\mathbf{s}$ and the inducing variables $(\bar{\mathbf{s}}, \bar{\mathbf{g}})$ is formulated based on the conditional distribution $p(\mathbf{g}|\mathbf{s}, \bar{\mathbf{s}}, \bar{\mathbf{g}}) = \int p(\mathbf{g}|\mathbf{s}, \mathbf{f})p(\mathbf{f}|\bar{\mathbf{s}}, \bar{\mathbf{g}})d\mathbf{f}$, where $\mathbf{f}$ are latent function values. A common approximation (used by Titsias (2009), related to FITC by Snelson & Ghahramani (2005)) leads to:

$$p(\mathbf{g}|\mathbf{s}, \bar{\mathcal{D}}, \bar{\mathbf{g}}) \approx \mathcal{N}\left(\mathbf{g}|\mathbf{k}_\mathbf{s}^\top \mathbf{K}_M^{-1}\bar{\mathbf{g}}, K_{\mathbf{ss}} - \mathbf{k}_\mathbf{s}^\top \mathbf{K}_M^{-1}\mathbf{k}_\mathbf{s} + \sigma^2\right), \tag{35}$$

where $[\mathbf{k}_\mathbf{s}]_m = K(\mathbf{s}, \bar{\mathbf{s}}_m)$ and $K_{\mathbf{ss}} = K(\mathbf{s}, \mathbf{s})$.

The overall likelihood for the set of observed subgoals $\mathbf{g}$ given corresponding states $\mathbf{s}$ and the inducing variables is approximated as:

$$\begin{aligned} p(\mathbf{g}|\mathbf{s}, \bar{\mathbf{s}}, \bar{\mathbf{g}}) &\approx \prod_{n=1}^{N} p(\mathbf{g}_n|\mathbf{s}_n, \bar{\mathbf{s}}, \bar{\mathbf{g}}) \\ &\approx \mathcal{N}\left(\mathbf{g} \mid \mathbf{K}_{NM}\mathbf{K}_M^{-1}\bar{\mathbf{g}}, \Lambda + \sigma^2\mathbf{I}\right). \end{aligned} \tag{36}$$

where $\Lambda$ is a diagonal matrix with $[\Lambda]_{nn} = K(\mathbf{s}_n, \mathbf{s}_n) - \mathbf{k}_n^\top \mathbf{K}_M^{-1}\mathbf{k}_n$, and $\mathbf{k}_n$ is the vector of kernel evaluations between $\mathbf{s}_n$ and all inducing states $\bar{\mathbf{s}}$. Note that $[\mathbf{K}_{NM}]_{nm} = K(\mathbf{s}_n, \bar{\mathbf{s}}_m)$.

A standard Gaussian prior is placed on the imaginary subgoals $\bar{\mathbf{g}}$:

$$p(\bar{\mathbf{g}}|\bar{\mathbf{s}}) = \mathcal{N}(\bar{\mathbf{g}}|0, \mathbf{K}_M). \tag{37}$$

Using Bayes' rule on Eq. 36 and Eq. 37, the posterior distribution over the imaginary subgoals $\bar{\mathbf{g}}$ given the observed data $\mathbf{D}_h = (\mathbf{s}, \mathbf{g})$ and inducing states $\bar{\mathbf{s}}$ can be derived as:

$$\begin{aligned} p(\bar{\mathbf{g}}|\mathbf{D}_h, \bar{\mathbf{s}}) &\propto p(\mathbf{g}|\mathbf{s}, \bar{\mathbf{s}}, \bar{\mathbf{g}})p(\bar{\mathbf{g}}|\bar{\mathbf{s}}) \\ &= \mathcal{N}\big(\bar{\mathbf{g}} \mid \mathbf{K}_M\mathbf{Q}_M^{-1}\mathbf{K}_{MN}(\Lambda + \sigma^2\mathbf{I})^{-1}\mathbf{g}, \\ &\qquad \mathbf{K}_M\mathbf{Q}_M^{-1}\mathbf{K}_M\big). \end{aligned} \tag{38}$$

where $\mathbf{Q}_M = \mathbf{K}_M + \mathbf{K}_{MN}(\Lambda + \sigma^2\mathbf{I})^{-1}\mathbf{K}_{NM}$. Note the inversion now involves $M \times M$ matrices, making computation feasible ($\mathcal{O}(NM^2)$ or $\mathcal{O}(M^3)$ depending on the exact method).

Finally, given a new state $\mathbf{s}_*$, the predictive distribution of the corresponding subgoal $\mathbf{g}_*$ is obtained by integrating the approximate likelihood (Eq. 35 applied to $\mathbf{s}_*$) against the posterior (Eq. 38):

$$\begin{aligned} p(\mathbf{g}_*|\mathbf{s}_*, \mathbf{D}_h, \bar{\mathbf{s}}) &= \int d\bar{\mathbf{g}}\, p(\mathbf{g}_*|\mathbf{s}_*, \bar{\mathbf{s}}, \bar{\mathbf{g}})\, p(\bar{\mathbf{g}}|\mathbf{D}_h, \bar{\mathbf{s}}) \\ &= \mathcal{N}\big(\mathbf{g}_* \mid \mu_*, \sigma_*^2\big). \end{aligned} \tag{39}$$

where the predictive mean $\mu_*$ and variance $\sigma_*^2$ are:

$$\mu_* = \mathbf{k}_*^\top \mathbf{Q}_M^{-1}\mathbf{K}_{MN}(\Lambda + \sigma^2\mathbf{I})^{-1}\mathbf{g} \tag{40}$$

and

$$\sigma_*^2 = K_{**} - \mathbf{k}_*^\top(\mathbf{K}_M^{-1} - \mathbf{Q}_M^{-1})\mathbf{k}_* + \sigma^2. \tag{41}$$

Here, $\mathbf{k}_*$ is the vector $[\mathbf{k}_*]_m = K(\mathbf{s}_*, \bar{\mathbf{s}}_m)$ and $K_{**} = K(\mathbf{s}_*, \mathbf{s}_*)$.

The inducing states $\bar{\mathbf{s}}$ and the GP hyperparameters $\theta_{gp} = \{\gamma, \ell, \sigma\}$ are typically optimized by maximizing the marginal likelihood (evidence) $p(\mathbf{g}|\mathbf{s}, \bar{\mathbf{s}})$, which can also be computed efficiently using the inducing points. This optimization encourages the inducing states to adaptively summarize the data distribution in the state space.

**Algorithm 1** HIDI

1: **Input:** Higher-level actor $\pi_{\theta_h}^h$, lower-level actor $\pi_{\theta_l}^l$, critics $Q^h$, $Q^l$, goal transition $h(\cdot)$, higher-level frequency $k$, training episodes $N$.
2: **for** $n = 1$ **to** $N$ **do**
3:     Sample initial state $s_0$
4:     $t \leftarrow 0$
5:     **repeat**
6:       **if** $t \equiv 0 \,(\mathrm{mod}\ k)$ **then**
7:         Sample subgoal via Eq. 15
8:       **else**
9:         Compute $g_t = h(g_{t-1}, s_{t-1}, s_t)$
10:       **end if**
11:       Sample action $a_t \sim \pi_{\theta_l}^l(a|s_t, g_t)$
12:       Execute $a_t$, observe $s_{t+1} \sim \mathcal{P}(s|s_t, a_t)$
13:       Compute intrinsic reward $r_t \sim \mathcal{R}(r|s_t, g_t, a_t)$
14:       Store transition $(s_{t-1}, g_{t-1}, a_t, r_t, s_t, g_t)$
15:       Sample `done` signal
16:       $t \leftarrow t + 1$
17:     **until** `done` is `true`
18:     **if** Training higher-level policy $\pi_{\theta_h}^h$ **then**
19:       Sample relabeled experience $(s_t, \tilde{g}_t, \sum r_{t:t+k-1}, s_{t+k})$
20:       Update $\pi_{\theta_h}^h$ and GP hyperparameters via Eq. 4
21:       Update critic $Q^h$
22:     **end if**
23:     **if** Training low-level policy $\pi_{\theta_l}^l$ **then**
24:       Sample experience $(s_t, a_t, g_t, r_t, s_{t+1})$
25:       Update $\pi_{\theta_l}^l$ and critic $Q^l$
26:     **end if**
27: **end for**

## A.4. Implementation

We implement the two-layer hierarchical policy network following the architecture of the HRAC (Zhang et al., 2020), which uses TD3 (Fujimoto et al., 2018) as the foundational RL algorithm for both the high and low levels.

Following the parameterization of Ho et al. (2020), the diffusion policies are implemented as an MLP-based conditional diffusion model, which is a residual model, *i.e.*, $\epsilon_\theta(g^i, s, i)$ and $\epsilon_\theta(a^i, s, g, i)$ respectively, where $i$ is the previous diffusion time step, $s$ is the state condition, and $g$ is the subgoal condition. $\epsilon_\theta$ is implemented as a 3-layer MLP with 256 hidden units. The inputs to $\epsilon_\theta$ comprise the concatenated elements of either the low-level or high-level action from the previous diffusion step, the current state, the sinusoidal positional embedding of time step $i$, and the current subgoal if it is a high-level policy. We provide further implementation details used for our experiments in Table 3.

## A.5. Algorithm

We provide Algorithm 1 to show the training procedure of HIDI.

## A.6. Environments

Our experimental evaluation encompasses a diverse set of long-horizon continuous control tasks facilitated by the MuJoCo simulator (Todorov et al., 2012), which are widely adopted in the HRL community. The environments selected for testing our framework include:

- **Reacher:** This task entails utilizing a robotic arm to reach a specified target position with its end-effector.

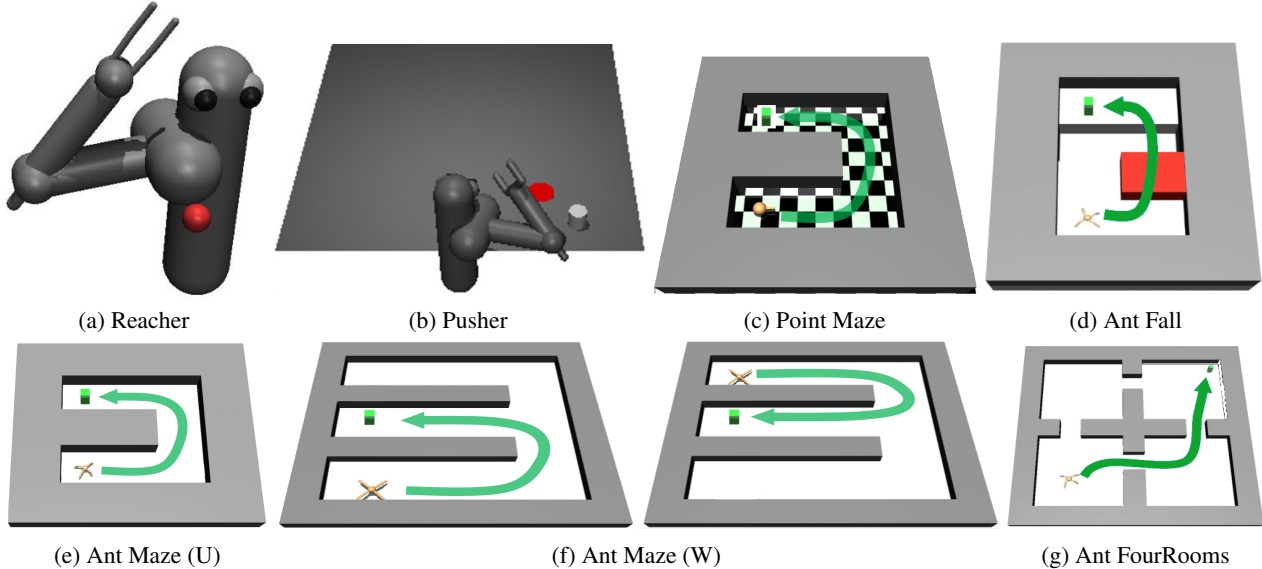

(a) Reacher    (b) Pusher    (c) Point Maze    (d) Ant Fall

(e) Ant Maze (U)    (f) Ant Maze (W)    (g) Ant FourRooms

*Figure 3.* Environments used in our experiments.

- **Pusher:** A robotic arm must push a puck-shaped object on a plane to a designated goal position.

- **Point Maze:** A simulation ball starts in the bottom left corner of a " ⊃"-shaped maze, aiming to reach the top left corner.

- **Ant Maze (U-shape):** A simulated ant starts in the bottom left of a " ⊃"-shaped maze, targeting the top left corner.

- **Ant Maze (W-shape):** A simulated ant starts at a random position within a "∃"-shaped maze, aiming for the middle left corner.

- **Ant Fall**: This task introduces three-dimensional navigation. The agent begins on a platform elevated by four units, with the target positioned across a chasm that it cannot traverse unaided. To reach the target, the agent must push a block into the chasm and then ascend onto it before proceeding to the target location.

- **Ant FourRooms**: In this task, the agent must navigate from one room to another to achieve an external goal. The environment features an expanded maze structure measuring $18 \times 18$.

- **Variants**: Following (Zhang et al., 2020; Kim et al., 2021), we adopt a variant with environmental stochasticity for Ant Maze (U-shape), Ant Fall, and Ant FourRooms by adding Gaussian noise with a standard deviation of $\sigma = 0.05$ to the $(x, y)$ positions of the ant robot at each step. Additionally, we adopt another variant labeled 'Image' for the large-scale Ant FourRooms environment. In this variant, observations are low-resolution images formed by zeroing out the $(x, y)$ coordinates and appending a $5 \times 5 \times 3$ top-down view of the environment, as described in (Nachum et al., 2019; Li et al., 2021).

We evaluate HIDI and all the baselines under two reward shaping paradigms: dense and sparse. In the dense setting, rewards are computed as the negative L2 distance from the current state to the target position within the goal space, whilst the sparse rewards are set to 0 for distances to the target below a certain threshold, otherwise -1. Maze tasks adopt a 2-dimensional goal space for the agent's $(x, y)$ position, adhering to existing works (Zhang et al., 2020; Kim et al., 2021). For Reacher, a 3-dimensional goal space is utilized to represent the end-effector's $(x, y, z)$ position, while Pusher employs a 6-dimensional space, including the 3D position of the object. Relative subgoal scheme is applied in Maze tasks, with absolute scheme used for Reacher and Pusher.

|   | HIDI | SAGA | HIGL | HRAC | HIRO |
|---|------|------|------|------|------|
| $\Delta$ | **0.82±0.08** | 0.95±0.13 | 1.57±0.05 | 1.72±0.06 | 10.90± 2.04 |

*Table 1.* The distance between generated subgoals and the reached subgoals, *i.e.*, the final state of k-step low-level roll-out, averaged over 10 randomly seeded trials with standard error.

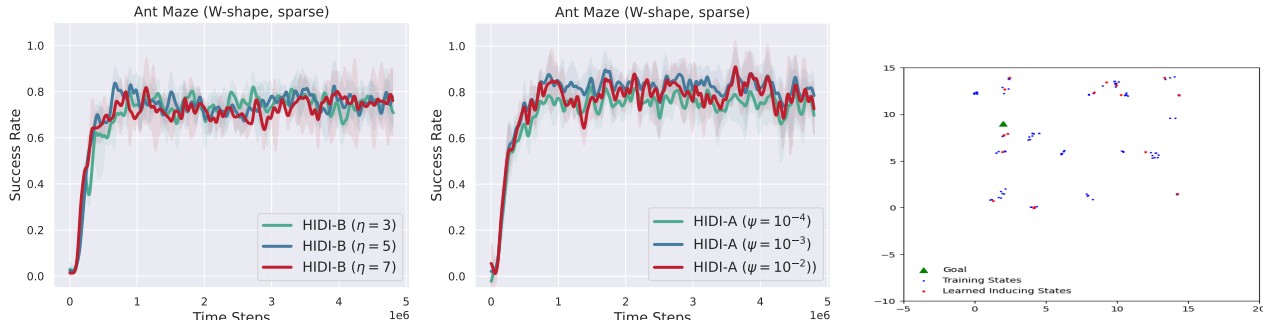

*Figure 4.* (Left) Impact of $\eta$, which balances the diffusion objective and RL objective. (Middle) Impact of $\psi$, which adjusts the influence of GP prior in learning the distribution of diffusional subgoals. (Right) Visualization of the learned inducing states (2D coordinates) compared with the complete training data.

## A.7. Additional Results

Figure 5 shows the generated subgoals and reached subgoals of HIDI and compared baselines in Ant Maze (W-shape, sparse) with the same starting location. HIDI generates reasonable subgoals for the lower level to accomplish, as evidenced by the minimal divergence between the generated and achieved subgoals which in turn provides a stable learning signal for the low-level policy. In contrast, the subgoals generated by HIRO are unachievable and cannot guide the agent towards the final target. Subgoals from HRAC frequently get stuck in local minima due to its local adjacency constraint. While HIGL and SAGA demonstrate improvement in constraining subgoals locally while escaping local optima, the high-level policy is inadequately compatible with the low-level skills. Specifically, the increasing gap between the generated subgoal and reached subgoal leads to inferior performance compared to HIDI.

We qualitatively study the learned inducing states in Fig. 4 (Right), where the states of the complete set of training batch (100) and the final learned inducing states (16) are visualized. Note, only the $x, y$ coordinates are selected from the state space in both cases for visualization purpose. We can observe that with a significantly smaller number of training data, the inducing points capture the gist of the complete training data by adapting to cover more critical regions of the state space, *e.g.*, the turning points in the Ant Maze (W-shape) task.

|  | Tasks | HIDI | HLPS | SAGA | HIGL | HRAC | HIRO |
|---|---|---|---|---|---|---|---|
| **Sparse** | Reacher | **0.95±0.01** | 0.54±0.14 | 0.80±0.05 | 0.89±0.01 | 0.74±0.10 | 0.76±0.02 |
|  | Pusher | **0.78±0.02** | 0.62±0.12 | 0.20±0.12 | 0.31±0.10 | 0.17±0.06 | 0.19±0.06 |
|  | Point Maze | **1.00±0.00** | 0.99±0.00 | 0.99±0.00 | 0.71±0.20 | 0.99±0.00 | 0.89±0.10 |
|  | Ant Maze (U) | **0.91±0.03** | 0.83±0.02 | 0.82±0.02 | 0.52±0.05 | 0.80±0.03 | 0.72±0.03 |
|  | Ant Maze (W) | **0.87±0.02** | 0.83±0.02 | 0.70±0.03 | 0.70±0.03 | 0.51±0.17 | 0.59±0.03 |
|  | Stoch. Ant Maze (U) | **0.91±0.02** | 0.80±0.05 | 0.81±0.01 | 0.75±0.03 | 0.73±0.02 | 0.61±0.03 |
|  | Stoch. Ant Fall | **0.84±0.02** | 0.78±0.04 | 0.48±0.05 | 0.52±0.02 | 0.25±0.03 | 0.28±0.03 |
|  | Stoch. Ant FourRooms (Img.) | **0.64±0.03** | 0.55±0.05 | 0.32±0.03 | 0.31±0.02 | 0.00±0.00 | 0.00±0.00 |
| **Dense** | Point Maze | **1.00±0.00** | 1.00±0.00 | 0.94±0.04 | 0.98±0.02 | 0.99±0.00 | 0.81±0.19 |
|  | Ant Maze (U) | **0.88±0.01** | 0.83±0.01 | 0.80±0.04 | 0.83±0.07 | 0.76±0.04 | 0.75±0.07 |
|  | Ant Maze (W) | **0.88±0.05** | 0.80±0.02 | 0.68±0.03 | 0.78±0.04 | 0.75±0.07 | 0.50±0.04 |
|  | Stoch. Ant Maze (U) | **0.92±0.01** | 0.88±0.03 | 0.87±0.03 | 0.82±0.03 | 0.70±0.04 | 0.80±0.03 |

*Table 2.* Final performance of the policy obtained after 5M steps of training with sparse rewards, averaged over 10 randomly seeded trials with standard error.

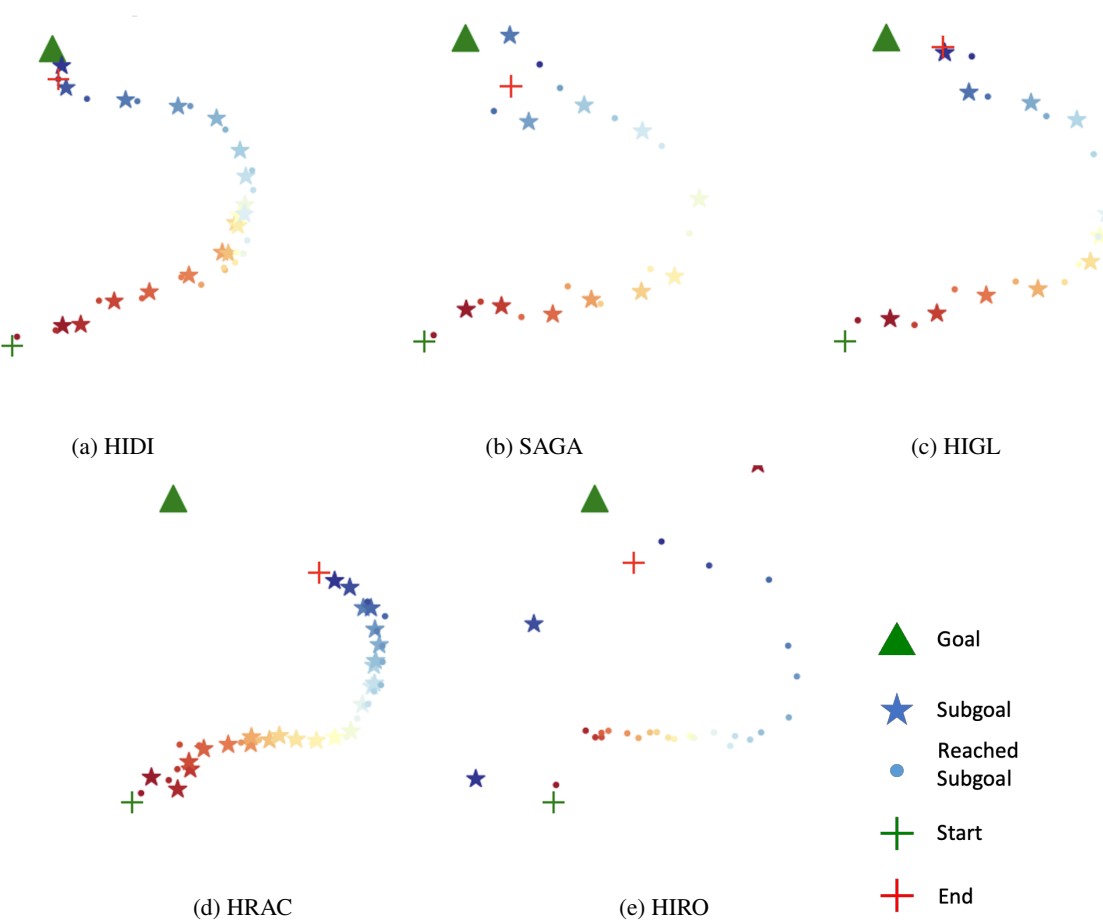

*Figure 5.* Visualization of generated subgoals and reached subgoals of HIDI and compared baselines in Ant Maze (W-shape, sparse) with the same starting location.

| Module | Parameter | Value |
|---|---|---|
| Diffusional Policy | Number of hidden layers | 1 |
| | Number of hidden units | 256 |
| | Nonlinearity, | Mish |
| | Optimizer | Adam |
| | Learning rate | $10^{-4}$ |
| | Hyperparameter for RL objective $\eta$ | 5 |
| | Number of diffusion steps $N$ | 5 |
| | GP loss weight | $10^{-3}$ |
| | GP learning rate | $3 \times 10^{-4}$ |
| | Subgoal selection probability $\varepsilon$ | 0.1 |
| | Number of inducing states | 16 |
| Two-layer HRL, critic networks | Number of hidden layers | 1 |
| | Number of hidden units per layer | 300 |
| | Nonlinearity | ReLU |
| | Optimizer | Adam |
| | Learning rate, critic | $10^{-3}$ |
| | Batch size, high level | 100 |
| | Batch size, low level | 128 |
| | Replay buffer size | $2 \times 10^5$ |
| | Random time steps | $5 \times 5^6$ |
| | Subgoal frequency | 10 |
| | Reward scaling, high level | 0.1 |
| | Reward scaling, low level | 1.0 |

*Table 3.* Network architecture and key hyperparameters of HIDI

