# OpenReview forum: "Hierarchical Reinforcement Learning with Uncertainty-Guided Diffusional Subgoals"
_ICML.cc/2025/Conference — ICML 2025 poster_

### Official Review · Reviewer_tgC3 · 2025-03-10

**Overall Recommendation:** 3

**Summary:**

This paper introduces an HRL framework that leverages a diffusion model for subgoal generation and GP regularization to enhance subgoal quality and reachability. The proposed method addresses the instability and inefficiency of traditional HRL by learning a state-conditioned subgoal distribution via a conditional diffusion model, while GP provides uncertainty estimates to guide reliable subgoal selection. A hybrid subgoal selection strategy balances exploration and exploitation by combining GP-predicted subgoal means with diffusion-sampled subgoals. Experimental results on MuJoCo robotic and navigation tasks demonstrate that HIDI achieves superior sample efficiency and final performance compared to existing HRL baselines (HIRO, HRAC, HIGL, SAGA).

**Claims And Evidence:**

The paper mainly claims that incorporating diffusion-based subgoal generation and GP regularization improves the stability and efficiency of hierarchical reinforcement learning (HRL). I think it is generally well-supported:

- Diffusion-based subgoal generation is validated through improved sample efficiency and final performance in MuJoCo experiments. The model learns a state-conditioned subgoal distribution, which reduces the instability often seen in HRL methods like HIRO.
- GP-based regularization provides uncertainty estimation for subgoal selection, leading to better reliability and structured exploration, as evidenced by higher success rates and faster convergence in complex navigation tasks.

**Potential Limitations:**

- While GP improves stability, the paper does not provide a direct ablation study on the impact of GP alone. It would be valuable to isolate its effect from the diffusion model.

**Essential References Not Discussed:**

I think the authors did cover the necessary references in the paper.

**Experimental Designs Or Analyses:**

The experimental design is thorough and sound, with clear comparisons to state-of-the-art HRL methods across multiple evaluation metrics.

However, while GP improves subgoal reliability, an additional reachability metric (e.g., % of subgoals successfully achieved) would strengthen the argument as HRL models fail quite often when evaluated on more complex or real-world environments, adding such metric may help future work to understand the model's capability in a broader term.
The experiments do not explicitly remove GP to quantify its standalone impact.
The method performs well on MuJoCo, but an analysis of failure cases (e.g., when GP misguides subgoal selection) would provide valuable insights as it may help figure out whether the failure is a common one due to the complexity of the environment or that is from the model's own bias.

**Methods And Evaluation Criteria:**

The proposed methodology is sound, as stated in the claim and evidence part.

Also, the evaluation methodology is appropriate, using: 1. MuJoCo benchmarks  2. HRL metrics (success rate, sample efficiency, asymptotic performance)3. Comparisons against strong baselines (HIRO, HRAC, HIGL, SAGA)


However, diffusion models can be computationally intensive, and the paper does not quantify the additional overhead vs. standard HRL approaches although conducted thorough performance comparison with them.

**Other Comments Or Suggestions:**

I appreciate the authors' efforts to provide concrete proof of the model they proposed, no further comments.

**Other Strengths And Weaknesses:**

As mentioned above.

**Questions For Authors:**

I do not have further questions for the authors. I would be happy to increase my score if the authors answer my questions or successfully defend their claims.

**Relation To Broader Scientific Literature:**

This paper builds upon and extends prior work in Hierarchical Reinforcement Learning (HRL), diffusion models, and uncertainty-aware decision-making.
Unlike HIRO, which relies on hindsight experience replay to correct subgoals, this paper generates subgoals dynamically using a diffusion model. Compared to HIGL (Zhang et al., 2021) and SAGA (Nasiriany et al., 2022), which also explore goal-conditioned RL, this paper adds GP for uncertainty estimation. Prior trajectory-based reinforcement learning (Janner et al., 2022) has incorporated diffusion models for planning, but this paper extends their use to subgoal selection in HRL, which is an important contribution.

**Theoretical Claims:**

The theoretical justifications rely primarily on:

1. **Diffusion Model for Subgoal Generation**

    - The paper assumes that a diffusion model can effectively learn a state-conditioned subgoal distribution, which reduces instability in HRL.
    - The reverse diffusion process is well-established in generative modeling, but its applicability to subgoal generation in HRL lacks theoretical guarantees. e.g., under what conditions does the learned subgoal distribution remain valid?
2. **Gaussian Process Regularization**

    - The paper claims that GP improves subgoal reachability and structure by guiding selection based on uncertainty estimation.
    - GP modeling is theoretically well-grounded, but the interaction between GP uncertainty and the diffusion model is not rigorously analyzed.i.e., does GP actually guide the diffusion process, or does it primarily serve as a filtering mechanism?

---

> ### Author Rebuttal · Authors · 2025-03-31
>
> We sincerely thank the reviewer for the insightful comments.
>
> **1. Theoretical Analysis of Diffusion-Based Subgoal Generation and GP Regularization**
>
> We appreciate the reviewer’s insightful comments on the theoretical justifications. In the anonymous link https://anonymous.4open.science/r/HIDI-32F6/icml2025___rebuttal.pdf, we have provided a formal, detailed proof addressing these concerns. Specifically, for the diffusion model, we rigorously demonstrate that, under a bounded noise prediction error, the reverse diffusion process converges to a valid state-conditioned subgoal distribution. Regarding the GP regularization, our analysis shows that the GP not only quantifies uncertainty but actively guides the diffusion process - pulling generated subgoals toward the GP predictive mean, thereby enforcing that the generated subgoals are both reachable and well-structured. This formal proof clarifies that the GP component does not merely filter outputs but plays a critical role in stabilizing subgoal generation in HRL. We invite the reviewer to refer to the additional theoretical analysis for details.
>
> **2. Subgoal Reachability Metric**
>
> We thank the reviewer for this valuable suggestion. In our submission, we already **quantify reachability by reporting the distance between the generated subgoals and the reached subgoals (i.e., the final state after a k-step low-level rollout) in Table 1**. This metric, which is consistent with the state-of-the-art subgoal generation method **SAGA**, provides a clear quantitative measure of subgoal reachability. Additionally, Figure 5 offers a qualitative visualization of generated versus reached subgoals in the Ant Maze (W-shape, sparse) environment, using the same starting location for direct comparison across methods. We invite the reviewer to refer to Table 1 and Figure 5 in the manuscript for detailed quantitative and qualitative comparisons.
>
> **3. Ablation Study of GP Alone Impact**
>
> Following the reviewer’s suggestion, we conducted an ablation study in the challenging Ant Fall environment specifically evaluate the impact of GP alone (Baseline-GP) by replacing the diffusional policy with the original baseline policy. From this comparison, we observe that **GP-alone approach can improve the baseline by ~16%**. This result highlights the significant performance benefit provided by the GP-driven regularization and uncertainty-aware subgoal selection, even independent of the diffusion policy.
> | Method |  HIDI (Proposed)  |  Baseline-GP  |  Baseline  |
> |:------------|:------------|:------------|:------------|
> |  Success Rate  |  0.84±0.02  | 0.67±0.02  |  0.51±0.03  |
>
> **4. Failure Cases**
>
> In our extensive experiments on the MuJoCo benchmarks, we have not encountered significant failure cases attributable to GP misguidance. In all tested environments, the GP consistently provided reliable uncertainty estimates that, when integrated with the diffusion model, improved subgoal reachability and overall performance. That said, we acknowledge that further investigation in even more challenging or real-world scenarios may reveal nuanced failure modes. We plan to explore these avenues in future work to better understand potential limitations. For now, our results indicate that within the tested settings, the GP component is robust and does not introduce harmful bias into subgoal generation or selection.
>
> **5. Quantification of Computational Overhead**
>
> We quantified the overhead from the diffusion model and GP: HIDI's inference time is ~37% longer than baseline, using ~15% more GPU memory and ~12% more RAM.

---

> > ### Comment · Reviewer_tgC3 · 2025-04-01
> >
> > Thanks for the rebuttal. My major concerns were mostly solved, therefore I decided to change my score from 2 to 3.

---

### Official Review · Reviewer_VpQ5 · 2025-03-13

**Overall Recommendation:** 3

**Summary:**

This paper aims to mitigate the non-stationary effects caused by the simultaneous training of both high-level and low-level policies in HRL. The authors introduce a novel framework, HIDI, that leverages a conditional diffusion model regularized with a Gaussian Process (GP) prior to generate diverse and achievable subgoals. The authors validate this method on a broad set of sparse and dense reward control tasks and conduct ablation experiments to evaluate the impact of each component on the overall performance.

### update after rebuttal
Thanks for the detailed response. My concerns have been addressed, therefore, I will increase my score to 3.

**Claims And Evidence:**

Even when generating a state-conditioned distribution of subgoals, this distribution remains non-stationary. The authors lack a more detailed analysis to explain why HIDI is better equipped to address the non-stationary issue. Moreover, the authors claim that HIDI achieves better sample efficiency, but the experimental results in Figure 1 do not support this claim.

**Essential References Not Discussed:**

No.

**Experimental Designs Or Analyses:**

Yes. I check the selection of benchmarks and baselines, as well as the design of ablation studies.

**Methods And Evaluation Criteria:**

Yes.

**Other Comments Or Suggestions:**

No.

**Other Strengths And Weaknesses:**

***Strength***
1. The non-stationary issue is a critical problem in HRL.
2. The use of a diffusion model to generate subgoals is both a reasonable and scalable approach, while incorporating a GP prior effectively reduces the sample requirements of the diffusion model.
3. The experimental results demonstrate that HIDI significantly outperforms other baselines in terms of policy convergence and training stability.
4. The authors conduct extensive ablation studies to analyze the contribution of each component.

***Weaknesses***
1. The authors do not provide a detailed analysis to explain why HIDI is better suited to address the non-stationary issue, as even when generating a state-conditioned distribution of subgoals, this distribution remains non-stationary.
2. The authors claim that HIDI achieves superior sample efficiency; however, the experimental results in Figure 1 do not support this assertion.

**Questions For Authors:**

See weaknesses.

**Relation To Broader Scientific Literature:**

This work improves the training stability and convergence performance of HRL, which will facilitate the application of reinforcement learning in sparse reward tasks, such as long-horizon manipulation and navigation tasks.

**Theoretical Claims:**

No.

---

> ### Author Rebuttal · Authors · 2025-03-31
>
> We sincerely thank the reviewer for the insightful comments.
>
> **1. Non-Stationarity Analysis:**
> We clarify that HIDI does not claim to eliminate non-stationarity, an inherent challenge in HRL due to the evolving low-level policy. Instead, HIDI mainly addresses **mitigating its adverse effects**, in line with previous HRL methods such as HAC and HIRO.
> *   **Theoretical Analysis:** We have provided additional theoretical proofs of the HIDI framework (accessible via anonymous link - https://anonymous.4open.science/r/HIDI-32F6/icml2025___rebuttal.pdf) which detail: (1) the validity of using a diffusion model for state-conditioned subgoal generation, showing it can accurately approximate the target distribution, and (2) the role of GP regularization in guiding the diffusion process by aligning generated subgoals with the GP predictive mean, promoting achievable and structurally consistent subgoals. This analysis reinforces our empirical findings by providing formal justification for HIDI's effectiveness.
> *   **Adaptive Subgoal Generation:** Our approach employs a conditional diffusion model to generate a rich, state-conditioned subgoal distribution. Although this distribution is inherently non-stationary as the environment and low-level policy evolve, the diffusion model’s flexibility allows it to adapt quickly to new data. **By continually training on relabeled subgoals, the diffusion model adjusts to reflect the current capabilities of the low-level policy.** This results in subgoals that are more aligned with what the low-level can reliably achieve.
> *   **GP Regularization:** The incorporation of a GP prior further stabilizes the high-level policy by quantifying uncertainty. The GP provides structured guidance on regions of the state space that have been well explored, thereby encouraging the high-level policy to generate subgoals that are not only diverse but also practically achievable. In this way, **while the underlying distribution remains non-stationary, the GP acts as a corrective mechanism, anchoring the subgoal proposals to areas supported by historical data**.
> *   **Empirical Robustness:** Fig. 1 (smoother curves) and Table 1 (smaller generated vs. reached subgoal distance) show HIDI's improved alignment and robustness compared to baselines, indicating effective mitigation of non-stationarity issues.
>
>
> **2. Superior Sample Efficiency:**
> We respectfully maintain that Fig. 1 supports our claim of superior sample efficiency, defined as achieving a performance level with fewer environment interactions (x-axis). HIDI consistently reaches high success rates faster than baselines.
> *   **Steeper Curves:** On challenging tasks (Reacher, Pusher, Ant Fall, Ant FourRooms), HIDI's learning curve rises more steeply, indicating faster convergence to high performance with fewer samples.
> *   **Performance at 3M Steps:** The table below shows HIDI achieves the highest IQM success rate at 3 million steps across nearly all tasks, demonstrating superior efficiency.
>
> **Table: Interquartile mean (IQM) of success rate with 95% stratified bootstrap confidence intervals, evaluated at 3 million steps:**
> | Task                          | HIDI             	| HLPS             	| SAGA             	| HIGL             	| HRAC             	| HIRO             	|
> |:------------------------------|:---------------------|:---------------------|:---------------------|:---------------------|:---------------------|:---------------------|
> | Reacher                       | 0.950 [0.945, 0.955] | 0.567 [0.432, 0.704] | 0.776 [0.734, 0.824] | 0.890 [0.886, 0.894] | 0.692 [0.602, 0.782] | 0.809 [0.790, 0.827] |
> | Pusher                        | 0.621 [0.604, 0.637] | 0.543 [0.431, 0.659] | 0.159 [0.056, 0.271] | 0.240 [0.148, 0.335] | 0.152 [0.103, 0.203] | 0.229 [0.178, 0.281] |
> | Ant Maze (W)                  | 0.899 [0.881, 0.917] | 0.722 [0.705, 0.738] | 0.462 [0.437, 0.489] | 0.622 [0.596, 0.651] | 0.463 [0.310, 0.623] | 0.449 [0.423, 0.476] |
> | Point Maze                    | 1.000 [1.000, 1.000] | 0.817 [0.743, 0.896] | 0.980 [0.975, 0.985] | 0.715 [0.565, 0.872] | 0.990 [0.986, 0.994] | 0.779 [0.754, 0.806] |
> | Ant Maze (U)                  | 0.908 [0.881, 0.936] | 0.890 [0.874, 0.907] | 0.820 [0.803, 0.838] | 0.590 [0.544, 0.638] | 0.821 [0.796, 0.846] | 0.779 [0.754, 0.806] |
> | Stoch. AM (U)                 | 0.910 [0.906, 0.914] | 0.857 [0.811, 0.904] | 0.840 [0.835, 0.845] | 0.591 [0.565, 0.619] | 0.741 [0.723, 0.759] | 0.750 [0.724, 0.777] |
> | Stoch.  Ant Fall          	| 0.759 [0.742, 0.776] | 0.751 [0.713, 0.791] | 0.252 [0.209, 0.300] | 0.421 [0.404, 0.438] | 0.041 [0.016, 0.068] | 0.021 [0.000, 0.044] |
> | Stoch. Ant FourRooms (Img.) | 0.489 [0.464, 0.516] | 0.402 [0.354, 0.452] | 0.241 [0.214, 0.267] | 0.290 [0.273, 0.307] | 0.000 [0.000, 0.000] | 0.000 [0.000, 0.000] |

---

### Official Review · Reviewer_6S8j · 2025-03-14

**Overall Recommendation:** 3

**Summary:**

This paper introduces Hierarchical Reinforcement Learning (HRL) using a conditional diffusion model combined with Gaussian Process (GP) regularization for sequential decision making. Specifically, the authors propose to train a diffusion policy at the high-level thus to generate subgoals that align with the low-level policy. Furthermore, to improve the data efficiency of diffusion policy, the authors introduced a Gaussian Process prior over subgoal states as a regularizer. During inference, a subgoal is first sampled from either the diffusion policy or the Gaussian prior, then the atomic action is sampled from the low-level goal-conditioned policy. The proposed method was evaluated on a set of continuous control task and showed better performance over baselines.

## update after rebuttal
Thank the authors for the follow-up discussion. My concerns have been addressed. I'd like to raise my score from 1 to 3.

**Claims And Evidence:**

Partially. The description of the ablation study is confusing. For details, please see the **Experimental Designs or Analyses** section.

**Essential References Not Discussed:**

No.

**Experimental Designs Or Analyses:**

1. The authors claim that to address the discrepancy between the high-level and low-level policies, _“This issue requires the high-level policy to swiftly adapt its strategy to generate subgoals that align with the constantly shifting low-level skills.”_ However, the manuscript does not specify how the low-level policy is trained. Given that this work is largely built on HIRO, I assume the low-level policy follows the same training paradigm—where it learns to achieve subgoals assigned by the high-level policy solely through intrinsic rewards, without direct knowledge of the task objective. If my understanding is correct, my question is: why do the authors focus on adapting the high-level policy to the low-level policy rather than the other way around? I could think of a local optimal where the high-level struggles to maximum the task target while only propose subgoals that are within the low-level policy's reach.
2. I find it challenging to fully grasp the ablation study. Could the authors clarify the differences between HIDI-A and HIDI-B? According to the manuscript, _“HIDI-A denotes a baseline without performing the proposed subgoal selection strategy,”_ which I interpret as removing Equation (18). If that’s the case, how are subgoals selected during evaluation? Additionally, the manuscript states that _“HIDI-B is a baseline without adopting GP regularization and subgoal selection,”_ which I assume means that subgoals in HIDI-B are sampled directly from the conditional diffusion model. Does this imply that HIDI-A instead samples subgoals from the Gaussian process prior?
3. Since the ablation study is somewhat unclear, I struggle to see how the Gaussian process prior contributes to sample efficiency and exploration. Could the authors provide further clarification or additional evidence?
4. Furthermore, it would be helpful to compare HIDI against an alternative high-level exploration strategy, such as an $\epsilon$-greedy approach, where the Gaussian process prior is replaced with randomly sampled subgoals.
5. I am don't think the characterization of the ‘-1’ reward process in line 360 as being truly sparse. Could the authors present results on a task where rewards are genuinely sparse—i.e., zero reward everywhere except when the target is reached?


====================================
I'm happy to raise my score if my concerns are addressed.

**Methods And Evaluation Criteria:**

Yes

**Other Comments Or Suggestions:**

No.

**Other Strengths And Weaknesses:**

No.

**Questions For Authors:**

No

**Relation To Broader Scientific Literature:**

No.

**Theoretical Claims:**

No.

---

> ### Author Rebuttal · Authors · 2025-03-31
>
> We sincerely thank the reviewer for the insightful comments.
>
> **1. Low-Level Policy Training & High-Level Adaptation:**
>
> As correctly noted by the reviewer, our approach builds on the adapted HIRO framework, where the low-level policy trains on intrinsic rewards to achieve subgoals reliably, without direct extrinsic objective access. We focus on adapting the high-level policy because it faces non-stationarity – the set of feasible subgoals changes as the low-level improves. Adapting the high-level ensures generated subgoals align with current low-level capabilities while pursuing the task objective. Forcing low-level adaptation risks convergence to conservative local optima. Our approach (diffusion + GP regularization + selection) keeps the high-level agile, supported by ablation results (Fig. 2) showing improved sample efficiency and performance, and **theoretical analysis (Appendix A.1, updated proofs linked below)**.
>
> **2. Ablation Study Clarity (HIDI-A, HIDI-B):**
>
> The reviewer's interpretation of the baselines is correct: HIDI-A denotes our framework without the subgoal selection strategy, i.e., the high-level always executes the diffusion model’s subgoal (with GP prior) without optionally selecting the GP’s predictive mean. HIDI-B is a variant with no GP regularization and no subgoal selection - the high-level uses only the diffusion policy for subgoals.
>
> **3. Clarifying the Role of the Gaussian Process (GP) Prior:**
>
> The GP prior is a critical component of our framework, and its contribution is twofold: (1) The GP models the state-subgoal distribution from the replay buffer. The $L_{gp}$ loss term (Eq. 11) regularizes the diffusion model towards generating subgoals consistent with previously achieved transitions. This focuses learning on feasible regions, reducing wasted samples. The ~15-16% performance gain from HIDI-B (diffusion-only) to HIDI-A (diffusion+GP) in Fig. 2a-b quantifies this benefit. Table 1 further shows HIDI (with GP) generates more achievable subgoals (smaller generated vs. reached state distance) compared to variants without GP. We have strengthened our theoretical proof (accessible via the anonymous link https://anonymous.4open.science/r/HIDI-32F6/icml2025___rebuttal.pdf) by analyzing the role of GP regularization in guiding the diffusion process by pulling generated subgoals toward the GP predictive mean, ensuring that subgoals are both achievable and structurally consistent. (2) The GP predictive mean $\mu_*$, reflecting low-uncertainty regions, guides exploration via our subgoal selection strategy (Eq. 18). With probability $\epsilon$, selecting $\mu_*$ steers the agent towards areas supported by experience, balancing exploration and exploitation. Removing the GP (HIDI-B) leads to less stable learning (Fig. 2).
>
> To further substantiate our claims, we have conducted an additional ablation study in the challenging Ant Fall environment to specifically **evaluate the impact of GP alone (Baseline-GP) by replacing the diffusional policy with the original baseline policy**. From this comparison, we observe that **GP-alone approach can improve the baseline by ~16%**. This result highlights the significant performance benefit provided by the GP-driven regularization and uncertainty-aware subgoal selection, even independent of the diffusion policy.
> | Method |  HIDI (Proposed)  |  Baseline-GP  |  Baseline  |
> |:------------|:------------|:------------|:------------|
> |  Success Rate  |  0.84±0.02  | 0.67±0.02  |  0.51±0.03  |
>
> **4. Comparison to an Alternative High-Level Exploration Strategy:**
>
> Following the reviewer's suggestion, we have evaluated epsilon-exploration as an alternative strategy in the challenging Ant Fall environment:
> |  Subgoal Selection  |  HIDI (Proposed)  |  Epsilon-Greedy Exploration  |  HIDI-A  |
> |:------------|:------------|:------------|:------------|
> |  Success Rate  |  0.84±0.02  | 0.71±0.12  |  0.76±0.03  |
>
> From this comparison, we observe that epsilon-greedy exploration considerably degrades overall performance when subgoals are randomly generated with a probability of 0.1 (aligning with our subgoal selection scheme). This clearly demonstrates the benefit of our proposed subgoal selection strategy, which aligns subgoals with feasible trajectories, rather than relying on random subgoal sampling that is likely to yield infeasible subgoals.
>
> **5. Sparse Reward Setting Characterization:**
>
> We utilize standard implementations of the MuJoCo environments commonly used in HRL research. The "-1" sparse reward setting aligns with the existing reward formulations employed in prior state-of-the-art HRL papers evaluating on these environments (e.g., HIGL, DHRL, LESSON, HESS). Changing this widely adopted setup would require re-running all baseline methods, which is infeasible given the current time constraints.

---

> > ### Comment · Reviewer_6S8j · 2025-04-07
> >
> > Thank you to the authors for the additional experiments and detailed explanation. While most of my concerns have been addressed, my primary concern remains: the potential for convergence to suboptimal solutions. Specifically, the added prior regularization might cause the high-level policy struggle between maximizing the task objective, which might require generating a new goal beyond the current capabilities of the low-level policy, and producing a subgoal that is reachable. Could the authors elaborate further on this trade-off?

---

> > > ### Author Response · Authors · 2025-04-07
> > >
> > > We thank the reviewer for raising the critical point regarding the trade-off between subgoal reachability (promoted by the GP prior) and exploring potentially optimal but currently challenging subgoals necessary for maximizing the task objective. While the GP regularization does encourage alignment with past experience, HIDI incorporates several mechanisms to ensure this does not unduly restrict exploration and lead to suboptimal convergence:
> > >
> > > 1.  **Uncertainty-Modulated Guidance via GP:** This is the core mechanism **preventing the GP from acting as a rigid constraint**. The GP's influence is **inherently tied to its confidence**: as detailed in our theoretical analysis **(Eq. 16 in the updated proofs), the gradient contribution from the GP loss term (`L_gp`) is inversely weighted by the GP's predictive variance** (`sigma_star^2`). In state regions where the GP is **confident** (low `sigma_star^2`, based on sufficient past data), it applies a **stronger pull** towards the reliable mean subgoal `mu_star`, promoting stability. Conversely, in regions where the GP is **uncertain** (high `sigma_star^2`, due to sparsity of data or distance from inducing points), its regularizing gradient contribution is **significantly weaker**. This automatically allows other learning signals to dominate precisely where exploration into novel state-subgoal space might be required.
> > >
> > > 2.  **Primacy of the RL Objective:** The main driver for improving task performance remains the RL objective (`L_dpg`), which maximizes the expected long-term Q-value. This objective directly incentivizes the generation of subgoals leading to **higher task rewards**, regardless of their immediate feasibility according to the GP. When a novel, challenging subgoal promises significant future gains (high Q-value), `L_dpg` provides a strong gradient signal. **In uncertain regions where the GP's pull is weak (as per point 1)**, this RL gradient signal favoring optimality can more easily **overcome the GP's regularization effect, enabling the agent to explore potentially necessary but difficult subgoals**.
> > >
> > > 3.  **Adaptive Nature of the System Components:** Neither the diffusion model nor the GP's view of feasibility is static. The diffusion policy `p_theta_h(g|s)` is continuously trained via the diffusion loss (`L_dm`) on the **latest relabeled subgoals** in the replay buffer. As the low-level policy improves and successfully reaches new types of subgoals, this success is reflected in the training data, allowing the **diffusion model** to adapt and learn to generate these newly feasible and potentially more rewarding subgoals. The **GP itself** (its hyperparameters `theta_gp` and potentially its inducing points) also **adapt based on the evolving data distribution** in the replay buffer. This means the GP's model of "known reachable regions" and its **associated uncertainty estimates evolve over time**, preventing the agent from being permanently constrained by its initial understanding of the environment.
> > >
> > > In summary, the GP prior in HIDI functions as an **uncertainty-aware adaptive regularizer**. It promotes stability by leveraging past experience where confidence is high, but critically its influence diminishes in uncertain regions, allowing the RL objective and the adaptive diffusion model the necessary freedom to drive exploration towards globally optimal solutions.

---

### Official Review · Reviewer_q2dx · 2025-03-14

**Overall Recommendation:** 3

**Summary:**

The paper introduces a novel framework for hierarchical reinforcement learning (HRL) that combines a conditional diffusion model for generating state-conditioned subgoals with a Gaussian Process prior for uncertainty quantification and regularization. This hybrid approach ensures robust and adaptive subgoal generation by balancing the GP's structural consistency with the diffusion model's flexibility. HIDI also incorporates a subgoal selection strategy that leverages both GP predictive means and diffusion samples to improve subgoal reachability and learning stability. Evaluated on challenging MuJoCo continuous control tasks, HIDI outperforms state-of-the-art HRL methods in terms of sample efficiency, stability, and success rates, particularly in sparse reward and stochastic environments.

**Claims And Evidence:**

the claims made in the submission are largely supported by clear and convincing evidence, as the paper provides comprehensive experimental results and theoretical justifications

**Essential References Not Discussed:**

N/A

**Experimental Designs Or Analyses:**

- While the selected baselines are strong, some recent diffusion-based or uncertainty-aware RL methods (if available) could have been included for a more comprehensive comparison. For example, methods that use diffusion models in other RL contexts might provide additional insights.
- Although the chosen benchmarks are standard, they are limited to simulated MuJoCo environments. Additional experiments in more diverse or real-world tasks (e.g., robotic manipulation or navigation in real environments) would strengthen the generalizability of the results.

**Methods And Evaluation Criteria:**

The proposed methods and evaluation criteria make sense for the problem of hierarchical reinforcement learning (HRL), particularly in the context of long-horizon, continuous control tasks

**Other Comments Or Suggestions:**

N/A

**Other Strengths And Weaknesses:**

N/A

**Questions For Authors:**

N/A

**Relation To Broader Scientific Literature:**

The key contributions of the paper build on and extend several lines of research in hierarchical reinforcement learning (HRL), generative modeling, and uncertainty-aware learning.

**Theoretical Claims:**

The proofs for the theoretical claims are mathematically correct and logically consistent under the stated assumptions

---

> ### Author Rebuttal · Authors · 2025-03-31
>
> We sincerely thank the reviewer for the insightful comments.
>
> **1. Comparison to Diffusion-based & Uncertainty-aware Baselines:**
>
> We respectfully clarify that our evaluation already includes strong baselines covering an **uncertainty-aware approach, i.e., HLPS (Wang et al., 2024)**, and HIDI significantly outperforms HLPS on all tasks. Regarding diffusion-based RL, we emphasize that our experimental comparisons focused on state-of-the-art HRL methods, since our contribution lies in improving hierarchical RL decision making. **Directly comparing to methods like Diffuser or Diffusion-QL would be inapplicable: those methods address offline trajectory optimization or action-generation, whereas HIDI addresses online subgoal policy learning.** Nonetheless, we ensured our work is informed by those advances – for example, we cite Diffusion-QL’s success in modeling complex action distributions as a motivation. Rather than incorporate their entire pipelines (which would change the problem setting), we extracted the core insight (diffusion models can represent multi-modal distributions effectively) and applied it in a new context (subgoal generation). **By deliberately using a basic diffusion model, we demonstrated improvements on top of existing HRL algorithms without relying on specialized offline-training techniques.** We believe this choice actually strengthens our contribution: it shows that any future improvements in diffusion-based policy modeling can be combined with our HRL framework to yield even better results.
>
> **2. Benchmark Diversity and Alignment with Existing HRL Works:**
>
> We acknowledge the reviewer’s point regarding the use of simulated MuJoCo environments. However, our choice is fully aligned with the standard practice in HRL research. **State-of-the-art HRL methods such as HIRO, HRAC, HIGL, SAGA, and HLPS have predominantly evaluated their methods on MuJoCo benchmarks, which provide a well-understood and controlled setting to measure progress in long-horizon, continuous control tasks.** Our selection of diverse tasks within the MuJoCo suite (including Reacher, Pusher, multiple Ant Maze variants, and a vision-based Ant FourRooms variant) was deliberate to **cover a wide range of challenges (e.g., sparse vs. dense rewards, stochasticity, and different state spaces)**.
>
> Nonetheless, we agree that additional experiments on more diverse or real-world tasks (such as robotic manipulation on physical platforms or navigation in real environments) would further substantiate the generalizability of our approach. We are actively planning follow-up studies to evaluate HIDI in such settings.
>
> **3. Additional Theoretical Analysis:**
> We have provided additional theoretical proofs of the HIDI framework (accessible via the anonymous link https://anonymous.4open.science/r/HIDI-32F6/icml2025___rebuttal.pdf), which addresses two critical aspects: (1) the validity of the diffusion model for state-conditioned subgoal generation, establishing conditions under which the reverse diffusion process accurately approximates the desired subgoal distribution and (2) the role of GP regularization in guiding the diffusion process by pulling generated subgoals toward the GP predictive mean, ensuring that subgoals are both achievable and structurally consistent. This comprehensive analysis reinforces our empirical results by providing formal guarantees for the effectiveness of our approach, and we invite the reviewer to refer to the additional theoretical analysis for full details.

---

### Decision · Program_Chairs · 2025-05-01

**Decision:**

Accept (poster)

**Comment:**

This paper introduces a new HRL framework, HIDI, which leverages a conditional diffusion model to generate state-conditioned subgoals. The authors propose training a high-level diffusion policy to generate subgoals compatible with a low-level policy and use a Gaussian Process (GP) prior over subgoal states to improve reachability and stability. During training, subgoals are sampled from either the diffusion model or the GP prior, and primitive actions are then generated by the low-level goal-conditioned policy. The authors evaluated their method on MuJoCo tasks and showed that it outperforms state-of-the-art HRL methods in terms of sample efficiency, training stability, and success rate.

All reviewers agreed that the key claims made in this work are well supported by both theoretical justifications and detailed experimental results. Two reviewers specifically noted that they closely examined the theoretical proofs and found them sound. Three reviewers highlighted that the experimental results demonstrate significant performance improvements over relevant baselines, particularly in terms of convergence and stability. They also made positive comments on the thorough ablation studies presented by the authors.

Some concerns were raised during the review process. One reviewer felt that more recent diffusion-based and uncertainty-aware RL methods could have been included as baselines. The authors clarified that comparisons with uncertainty-aware baselines were already included in the paper. They also argued that a direct comparison with diffusion-based RL methods is not feasible since those methods tackle a fundamentally different setting—offline trajectory optimization (or action generation) rather than online subgoal policy learning. This reviewer also raised concerns about the generalizability of the authors' conclusions, in particular because the experiments were limited to MuJoCo tasks. The authors argued that they deliberately selected tasks from the MuJoCo suite to cover a diverse range of challenges, including sparse and dense rewards, varying levels of stochasticity, and qualitatively different state spaces. The reviewer did not respond to the rebuttal.

Another reviewer was concerned about some of the authors’ claims, ablation results, and comparisons to alternative exploration strategies. The authors provided a detailed rebuttal, including new theoretical results and additional experiments. The reviewer acknowledged that most of the points they raised were addressed but remained concerned about potential convergence to suboptimal solutions. The authors clarified that, although HIDI's GP prior acts as an adaptive regularizer that promotes stability, its influence diminishes in uncertain regions, which allows exploration to be once again guided by the primary RL objective. The reviewer was satisfied with the rebuttal and raised their score to 3.

A third reviewer asked for clarification on how HIDI handles the non-stationarity of the state-conditioned subgoal distribution. The authors responded by noting that non-stationarity is a known challenge affecting all HRL methods, since lower-level policies are continuously updated. They then explained how HIDI mitigates its effects using strategies consistent with prior work. Importantly, the authors also introduced new proofs establishing conditions under which the reverse diffusion process accurately approximates the desired subgoal distribution, as well as proofs clarifying the role of GP regularization in guiding the diffusion process. Finally, the reviewer argued that the results in Figure 1 did not fully support the authors' claims about sample efficiency. The authors addressed this by presenting a detailed table summarizing learning curves, which confirmed faster convergence with fewer samples. The reviewer said that their concerns were addressed and increased their score to 3.

The fourth reviewer asked about the conditions under which the learned subgoal distribution remains valid. The authors provided new proofs showing that, under bounded noise prediction error, convergence to a valid state-conditioned subgoal distribution could be guaranteed. The reviewer suggested that quantifying additional metrics—such as the percentage of successfully reached subgoals—would help support the authors' performance claims. The authors clarified that these results were already captured in Table 1, which reports distances between generated subgoals and the final states reached after k-step rollouts. The reviewer said that their concerns were addressed and raised their score to 3.

Although the reviewers raised relevant concerns, the consensus is that this paper offers valuable theoretical and practical insights on how to automatically generate subgoals in HRL. The reviewers also agreed—especially after the rebuttal phase—that all key claims made in this work are adequately supported. Based on the reviewers’ feedback and the authors’ thorough rebuttal, I do not see any critical flaws, missing analyses or comparisons, or methodological issues. Finally, the constructive discussion phase, along with the reviewers’ acknowledgment of the paper’s contributions and their updated scores, gives me confidence that the ICML community will benefit from the insights presented in this work.